# Mixed effects of a national protected area network on terrestrial and freshwater biodiversity

Andrea Santangeli [1,2] ✉, Benjamin Weigel [1,3], Laura H. Antão [1], Elina Kaarlejärvi [1], Maria Hällfors[1,4], Aleksi Lehikoinen [5], Andreas Lindén[6], Maija Salemaa[6], Tiina Tonteri[6], Päivi Merilä [6], Kristiina Vuorio[4], Otso Ovaskainen[1,7,8], Jarno Vanhatalo [1,9], Tomas Roslin [1,10,11] & Marjo Saastamoinen [1,12]

Protected areas are considered fundamental to counter biodiversity loss. However, evidence for their effectiveness in averting local extinctions remains scarce and taxonomically biased. We employ a robust counterfactual multi-taxon approach to compare occupancy patterns of 638 species, including birds (150), mammals (23), plants (39) and phytoplankton (426) between protected and unprotected sites across four decades in Finland. We find mixed impacts of protected areas, with only a small proportion of species explicitly benefiting from protection—mainly through slower rates of decline inside protected areas. The benefits of protection are enhanced for larger protected areas and are traceable to when the sites were protected, but are mostly unrelated to species conservation status or traits (size, climatic niche and threat status). Our results suggest that the current protected area network can partly contribute to slow down declines in occupancy rates, but alone will not suffice to halt the biodiversity crisis. Efforts aimed at improving coverage, connectivity and management will be key to enhance the effectiveness of protected areas towards bending the curve of biodiversity loss.

The unprecedented scale of anthropogenic appropriation of nature is fueling biodiversity declines and ecosystem degradation, with severe repercussions for human health and wellbeing[1]. Protected areas (hereafter PAs) represent the main bastion for mitigating population declines and preventing extinctions[2]. However, this claim is often based on weak or sparse evidence[3–5], with studies reporting contrasting effects of PAs (e.g.[5–10]).

Evaluating PA performance is challenging, and most assessments use proxies for ecosystem conditions, such as forest loss or other metrics of anthropogenic pressure to measure the effect of

[1]Research Centre for Ecological Change, Organismal and Evolutionary Biology Research Programme, Faculty of Biological and Environmental Sciences, University of Helsinki, Helsinki, Finland. [2]Animal Demography and Ecology Unit, Institute for Mediterranean Studies (IMEDEA), CSIC-UIB, 07190 Esporles, Spain. [3]INRAE, EABX, Cestas, France. [4]Nature Solutions Unit, Finnish Environment Institute (SYKE), Helsinki, Finland. [5]Finnish Museum of Natural History, University of Helsinki, Helsinki, Finland. [6]Natural Resources Institute (LUKE), Helsinki, Finland. [7]Department of Biological and Environmental Science, University of Jyväskylä, P.O. Box 35 (Survontie 9C), FI-40014 Jyväskylä, Finland. [8]Centre for Biodiversity Dynamics, Department of Biology, Norwegian University of Science and Technology, Trondheim, Norway. [9]Department of Mathematics and Statistics, Faculty of Science, University of Helsinki, Helsinki, Finland. [10]Spatial Foodweb Ecology Group, Department of Agricultural Sciences, University of Helsinki, Helsinki, Finland. [11]Spatial Foodweb Ecology Group, Department of Ecology, Swedish University of Agricultural Sciences, Uppsala, Sweden. [12]Helsinki Institute of Life Science, University of Helsinki, Helsinki, Finland. ✉e-mail: andrea.santangeli@helsinki.fi

protection[10–12]. Studies using direct biodiversity measures, such as based on wildlife survey data, are scarce[8,13]. In addition, few studies have employed a counterfactual approach (sensu[3]), where the outcomes of area-based protection are compared to the outcomes in the absence of the intervention in otherwise similar areas[14]. Moreover, previous assessments of PA performance have also been taxonomically restricted, usually focusing on a single or a restricted group of species, and with biases towards well-studied or charismatic species (e.g. waterbirds[13]; carnivores[15]).

The lack of robust assessments of PA impacts is problematic, in particular given that none of the previously established global conservation targets have been fully met (Convention on Biological Diversity Aichi Targets[16]). What compromises many studies to date is the type of comparison adopted. Most studies have compared species performance between protected and unprotected sites, but neglecting that sites may differ in terms of a wealth of ecological and socioeconomic factors beyond their status as being protected or not, potentially leading to inadequate comparisons[3]. The current state of knowledge is particularly troubling in the light of management planning and policies for biodiversity conservation post 2020. While ambitious plans to further expand the global network of protected areas[17] have recently been formalized[18], the real impact of PAs as a conservation tool remains largely unquantified[12,19]. It is becoming clear that without a counterfactual approach, the effects of PAs in preserving biodiversity, as well as other ecosystem services and functions, may be overestimated[5,20], resulting in overoptimistic claims on their effectiveness. Thus, the need for stringent evaluation of which conservation measures are effective, and for which species, is more urgent than ever. In this context, counterfactual approaches can greatly improve our understanding of conservation effectiveness[3,5], and their use in conservation impact evaluation has been growing during recent years. In addition, the few cases where PA effectiveness has been evaluated across multiple taxonomic groups show mixed and taxon-specific impacts (e.g.[8]). Part of these apparent contrasts among taxa may perhaps be attributed to un-comparable protected and unprotected sites, with responses by different taxa being studied at different spatial scales[21,22]. Overall, this points to the need for assessments integrating three aspects: counterfactual approaches, the inclusion of multiple taxonomic groups, and a focus on patterns among taxa examined at the same spatial extent.

Here we take advantage of a unique set of wildlife survey data systematically collected in both protected and unprotected sites in Finland, containing over half a million observations. The data span over four decades, and cover taxa as diverse as birds, mammals, plants and freshwater phytoplankton (Fig. 1). We evaluated PA performance for 638 species by applying a counterfactual approach built on matching protected sites with comparable unprotected sites (see Methods). Given the latitudinal (stretching over 1100 km) and climatic range of Finland, as compared to relatively homogeneous socioeconomic conditions across the country, we focus on comparing sites with matching latitudinal, environmental and biogeographic features, as well as sites with similar anthropogenic footprints. As our key measure of PA outcome, we quantify changes in occurrence over time in relation to protection status using a joint species distribution modeling framework. Based on estimated occurrence trends, we classify species' response to PAs into those showing positive, negative and no detectable change over time compared to change in non-protected sites. Species' responses are further compared to their overall occurrence trend—i.e., we assess whether PAs have positive effects by either mitigating an overall decline, i.e., yielding lower rates of decline inside PAs, or by amplifying an overall increase, i.e., leading to steeper increases inside PAs. In addition, we investigated which factors may contribute to PA impact, by (1) evaluating if PA size, IUCN protection category and time since establishment affected PA effectiveness, and (2) assessing whether species responses to

protection were related to their traits, namely size, climatic niche and threat status.

## Results and discussion
### Overall impacts of protected areas
Using a robust analysis integrating site matching[13] with the hierarchical modeling of species communities[23], we unveiled mixed effects of protected areas on boreal biodiversity (Fig. 2). Overall, the occurrence of most species was unaffected by protection, showing similar trends in protected sites and in their unprotected counterparts (Fig. 2, Supplementary Fig. 1 and Supplementary Data 1). The overall lack of protection effect in the paired PA vs non-PA comparison was particularly evident for plants and phytoplankton, where most species showed no response to PAs. Among the species that did show a response to PAs, a varying proportion responded positively (3 to 19%) or negatively (5 to 12%; Supplementary Fig. 1) across all four groups. Almost one in five (19%) bird species showed a positive response to protection—i.e., their occurrence trend was more positive or less negative inside compared to outside of PAs, while 12% showed a negative response to protection —i.e., their occurrence trend was less positive or more negative inside compared to outside of PAs (Fig. 2 and Supplementary Fig. 1). A similar but weaker pattern emerged for mammals, with 13% of species responding positively and 9% negatively to protection. Among plant and phytoplankton, equally few species showed positive (3 and 4%, respectively) or negative (5 and 6%, respectively) responses (Fig. 2 and Supplementary Fig. 1).

Importantly, we found that PA size and time since protection enhanced the positive effects of PAs, with the magnitude of this improvement varying between the taxonomic groups. Specifically, focusing only on the sites within PAs larger than the median size, and their unprotected counterpart, we found that for mammals, plants and phytoplankton, the number of species responding positively to protection increased compared to when including all PAs regardless of size (to 22, 7 and 11% respectively), but not for birds (Supplementary Figs. 2, 3). In addition, focusing only on the years after a site was officially protected, we found that for birds and phytoplankton, the number of species responding positively to protection increased (to 25 and 14%, respectively), while negative responses were no longer detected for both birds and phytoplankton (Supplementary Figs. 4, 5). Conversely, mammal occurrence trends were unaffected by the timing of PA designation. On the other hand, we found no detectable effect of the IUCN PA category (i.e., the level of protection), on species' responses; specifically, the overall patterns of response were similar when re-running the analysis only for the PAs with the strictest protection level (Supplementary Figs. 6, 7).

Our results add to recent advances in counterfactual approaches for impact evaluation in conservation[14,20,24], which combined with large-scale and long-term data now allow to robustly quantify impacts of area-based protection on wildlife[13,15,25]. By applying these approaches to high-resolution systematic monitoring data from a wide range of taxa, we achieve a major advance in quantifying the impact of protection across the tree of life. Our findings of mixed responses to protection align with other recent studies[5,13,15], highlighting that protection can contribute to shaping species occurrence trajectories, but that the effects are highly species-specific and depend on key features of the protected areas, such as the timing of protection and the size of the PA.

### Species level responses to protection
We unveiled a wide distribution in the effect of protection on occurrence trends, ranging from strongly positive to strongly negative species-specific effects in all taxonomic groups (Fig. 2). Such PA effects were largely unrelated to species traits, such as whether a species is threatened or not, or how large it is (in terms of body size), or what temperatures it prefers (as reflected by the species' thermal index;

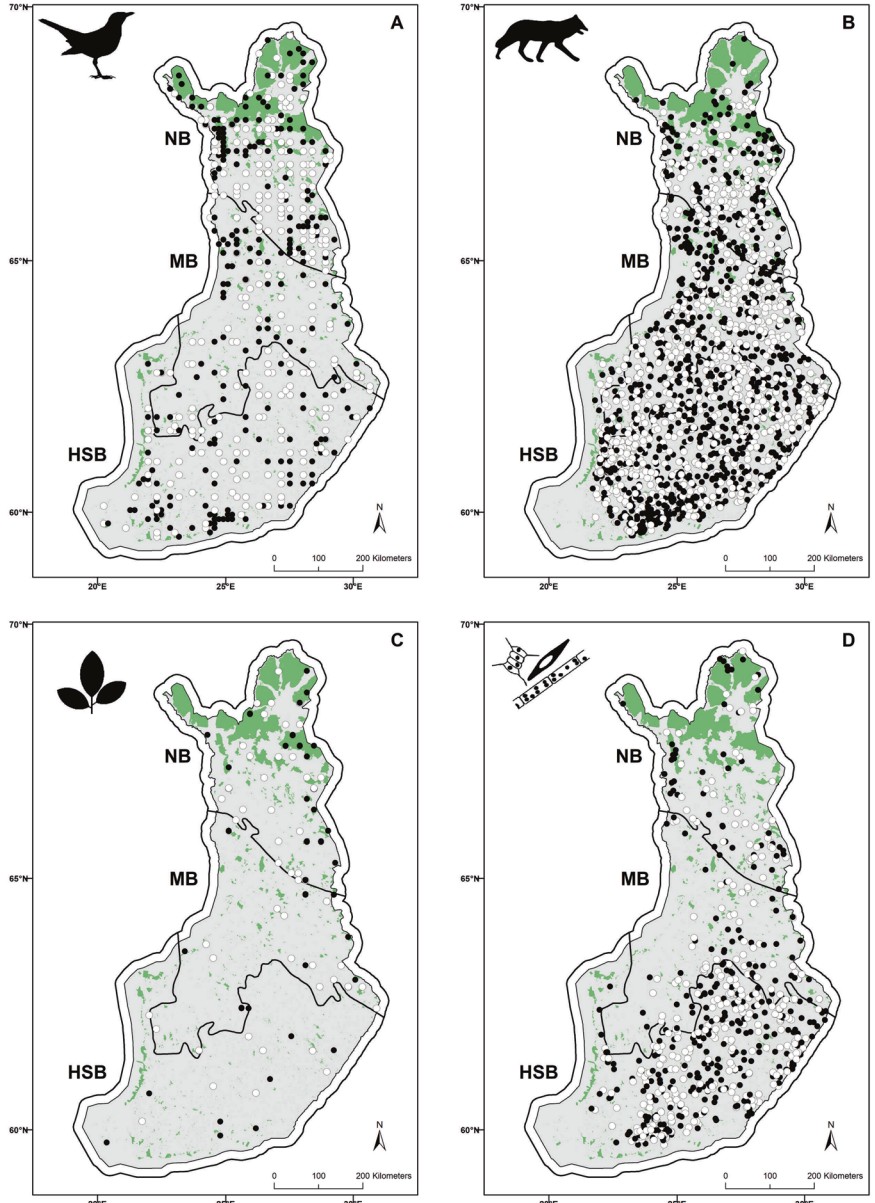

**Fig. 1 | Distribution of the sampling sites inside and outside of Protected Areas.** Protected (black circles) and unprotected (white circles) sites as identified from the matching approach for (**A**) birds, (**B**) mammals, (**C**) plants, and (**D**) phytoplankton. Protected areas are shown in green. Note that many protected areas, especially in the south, are too small to be visible at this resolution, and sites too close to each other may also be indistinguishable. The three vegetation zones used in the analysis are delimited by the thick black lines: HSB−Hemi- and South Boreal, MB−Middle Boreal, and NB−North Boreal zone. Coordinate scale labels refer to the Finnish EUREF FIN system.

Supplementary Table 1). The only exception was plants, for which we found evidence that warm-dwelling species showed more positive trends outside compared to inside of PAs ($p = 0.018$, Supplementary Table 1). Given the general pattern of a northward expansion of warm-affiliated species[26], this pattern suggests that protection may be preventing the colonization of Finnish PAs by warm-dwelling plants. We note that we were unable to analyze the effect of threat status for plants, or of threat status and thermal niche for phytoplankton (see Methods for details).

We further revealed the different ways by which PAs can result in positive or negative effects on species occurrences over time (Fig. 3). For birds, mammals, and phytoplankton, the positive impact of PAs mainly emerged by mitigating declines in occurrence; i.e., if species declined both inside and outside of protected areas, the rate of decline was slower in protected areas (yellow bars with a positive effect of PA in Fig. 3). Conversely, negative impacts of PAs were detected mostly through slower rates of increase inside PAs for species increasing both inside and outside of PAs (red bars on right side in Fig. 3). Plants showed more mixed responses, with some species having positive occurrence trends inside PAs but negative trends outside PAs (blue bars in Fig. 3), or vice versa−negative occurrence trends inside PAs but positive trends outside (grey bars in Fig. 3). Overall, species for which protection acted to reverse a trend occurring outside of PAs were exceptionally uncommon (blue and grey bars in Fig. 3).

The general finding of a limited effectiveness of PAs in reversing species decline may be attributed to the low coverage and connectivity of the Finnish PA network, which is likely too fragmented to counter biodiversity declines[27]. As a result, the only measurable impact of protection is manifested as a slowed-down rate of decline of species, as individuals likely find their last remnants of suitable habitat within PAs.

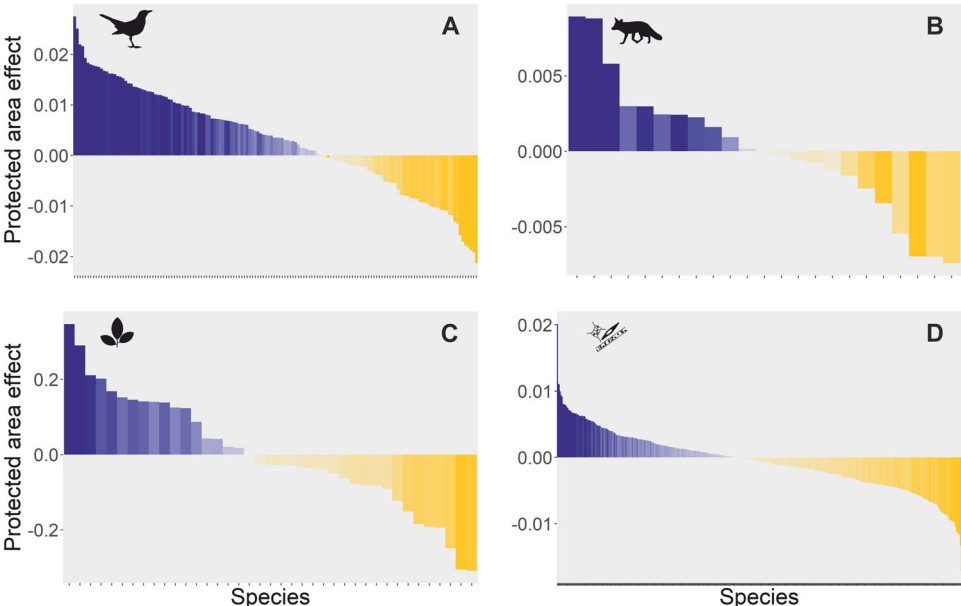

**Fig. 2 | Species-specific effects of protected areas.** The effect of protected areas on species occurrence trends, measured as the effect size of the interaction term between protection and year in the joint species distribution models for each taxonomic group: (**A**) birds, (**B**) mammals, (**C**) plants, and (**D**) phytoplankton. Each bar represents the effect of protection on a single species, from the most positive (blue) to the most negative effect (yellow), where darker colors indicate higher statistical support for the effect (posterior probability for the response). *X*-axes tick marks depict patterns of protected area effects for each species.

This interpretation is supported, at least partly, by patterns emerging from our results based only on the larger PAs (i.e., with size exceeding the median size across all protected area sites).

Our results underscore that when protection affected species occurrence trends, this typically occurred by either mitigating declines or hindering increases, while complete reversal of trends were rarely detected. Generally, cases where PAs actually mitigate or reverse declines provide strong evidence for positive conservation outcomes. This is because such effects will either delay or prevent local extinctions, thus allowing time for adaptation and dispersal of the species as well as more incisive and effective conservation strategies to be defined and implemented[28,29]. Such strategies could then aim to improve the connectivity and effectiveness of PA networks via improved management, with an increased focus on outcome-based targets[28,30] and on other effective area-based conservation measures[31]. Overall, the very diverse species responses to protection unveiled here underscore an additional challenge for the Kunming-Montreal Global Biodiversity Framework (GBF), aiming to protecting 30% of land and sea areas by 2030[17]—as biodiversity responses to protection are not universal, a single conservation target is unlikely to fit all taxa.

Importantly, increases and decreases in species incidence should be put into perspective. For instance, the apparent "negative" effect of protected areas in hindering increases (or colonizations), such as the case for warm-dwelling plants (see Supplementary Table 1), can be interpreted as a positive outcome at high northern latitudes[32,33]. They indicate that warm-dwelling plants may not easily colonize PAs, which represent important bastions for locally adapted cold-dwelling species with nowhere to move under climate change[32]. On the other hand, the PA effect of hindering increases of species occurrence may also be interpreted as negative, because it mitigates the reshuffling of communities to new conditions under global change. As such, it may lead to local or global extinctions[34]. Finally, the lack of systematic effects of protection found here should be interpreted not only in view of the present conditions, but also and foremost in light of the rapid global change in the near- and long-term. To this end, PAs can still be considered beneficial, as they ensure the preservation of habitats and ecosystems (not directly measured and assessed in this study) that

could otherwise be degraded or lost if human appropriation of nature continues unabated. We also caution that here we used only occurrence data, and future studies using abundance data may find more positive effects of PAs.

Overall, the effect of PAs was clearly influenced by the year in which the area was protected, and the size of the protected area. This adds to the evidence for genuine effects of protection. Thus, the lack of positive effects of protection for each and every species in our study should not be interpreted as if PAs are ineffective overall. Instead, it points to unequal benefits of protection for different species, and to the fact that area-based protection will be insufficient to act as a single silver bullet for countering species loss. Thus, the evidence provided here should be taken as a wake-up call that rigorous assessments of the effectiveness of protection are urgently needed, if we are to leverage the real potential of PAs in mitigating the global biodiversity crisis. Such evidence is needed across the tree of life, and should be incorporated in current management strategies. Ultimately, our findings suggest that by mitigating declines, PAs may allow time for multilateral environmental agreements, such as the Convention on Biological Diversity (CBD) post-2020 framework, to take effect before we reach a tipping point of biodiversity loss.

## Methods
### Wildlife survey data
To assess the effectiveness of PAs, we used data systematically collected within several monitoring schemes across Finland (see also[35]), amounting to over half a million observations. The data represent occurrence records from 1980 onwards for four different taxonomic groups: Birds (~141k observations, 1187 unique survey sites and 178 species), mammals (~131k observations, 2304 unique survey sites and 24 species), forest understory plants (hereafter plants; ~43k observations, 1712 unique survey sites and 350 species), and freshwater phytoplankton (~251k observations, 1836 unique survey sites and 886 species).

**Birds.** Bird data have been systematically collected using line transect censuses in Finland since the 1970s[36]. The data are collected annually

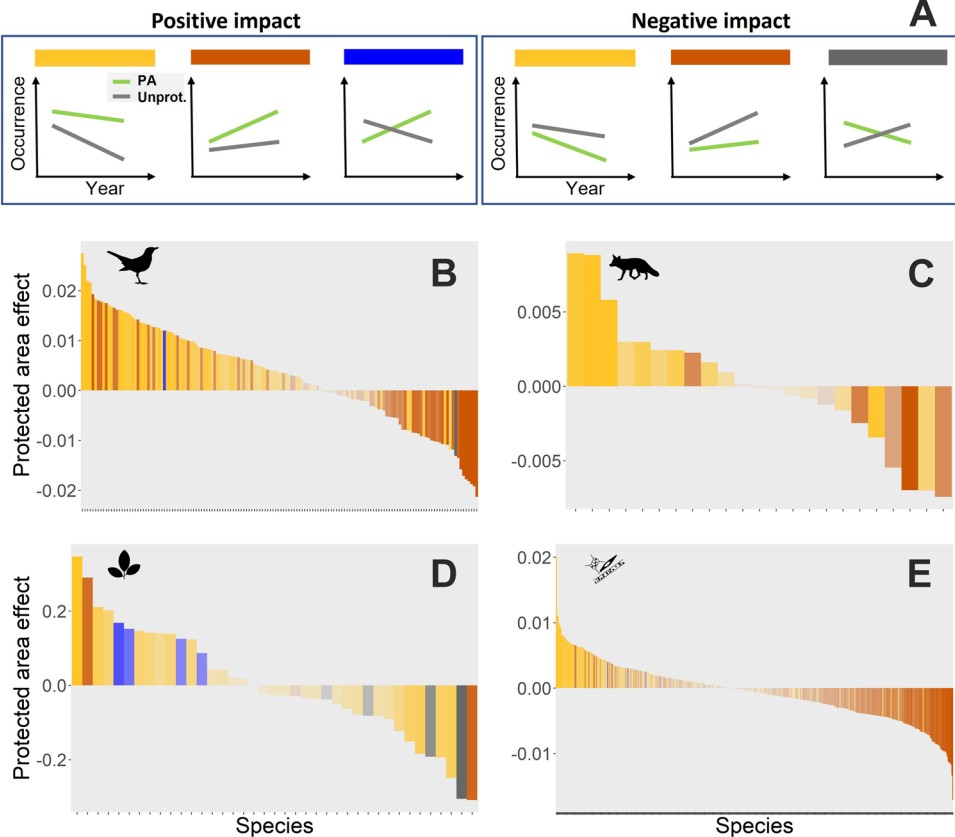

**Fig. 3 | Different means through which protected areas affected species occurrence trends. A** The six potential processes via which protected areas can impact species occurrence over time, where the green line represents protected areas and the grey line refers to unprotected areas. Positive impacts of protected areas can be observed through (1) alleviating rates of decline (yellow bars), (2) accelerating rates of increase (red bars), or (3) inverting negative occurrence trends occurring outside of protected areas to positive trends inside of protected areas (blue bars). Negative impacts can be observed through (4) accelerating the rate of decline (yellow bars), (5) alleviating the rate of increase (red bars) compared to unprotected areas, or (6) inverting occurrence trends from positive outside to negative inside protected areas (grey bars). Below we show the observed effects of protected areas on occurrence trends for each group: (**B**) birds, (**C**) mammals, (**D**) plants, and (**E**) phytoplankton. Such effects are estimated by the interaction between protection and year, from the most positive (left) to the most negative effect (right); darker colors indicate higher statistical support (see methods for details how support was defined) for the effect. Different bar colors (in **B**–**E**) match those on the top bars of the various protected area effects shown in (**A**). *X*-axes tick marks in (**B**–**E**) depict patterns of protected area effects for each species.

based on a one-visit census, in which birds are counted along a transect with a typical length of 3–6 km. The transects are pre-established (i.e., are assigned to known locations that are fixed over time), but not all transects are sampled every year. The census period is June, during the peak of the birds' breeding season, with observations typically carried out in dry weather conditions between 3:00 and 9:00 am, when the singing activity of birds is highest. The observer walks at a speed of 1.00–1.33 km/h (depending on the density of birds along the transect) using a map, compass, or global positioning system. The observations are carried out earlier in southern Finland (June 1–20) compared to northern Finland (June 10–30) due to later breeding phenology in northern latitudes. The line transect is divided into a main belt and a supplementary belt. The main belt is 50 m wide (25 m on each side of the transect line), and the supplementary belt represents the area beyond the main belt as far as birds can be detected. Every observation is assigned either to the main or the supplementary belt. Birds crossing the main belt belong to the supplementary belt even if first observed above the main belt. Species-specific annual proportions of observed birds and birds in the main belt remained stable between 1987–2010, indicating no major change in species detectability[37]. The data are curated by the Finnish Museum of Natural History. For this study, we used records collected between 1980 and 2019. The line transects were available as digitized lines in a Geographic Information System.

**Mammals**. A systematic monitoring program of counts of mammal snow tracks during the winter was established in 1989, as part of the wildlife triangle scheme (also referred to as game triangle data). The data are curated by the Natural Resources Institute Finland (Luke[38,39]); The scheme is based on a network of triangle-shaped 4 × 4 × 4 km transects (totaling 12 km per triangle), with fixed locations covering the entire country. Out of the total of over 2000 established triangles, about one third are surveyed annually. The triangles are located in forested areas covering the main forest types in Finland and the survey is carried out by volunteers (mainly hunters). Each triangle is surveyed for snow tracks within 1 day. All tracks of 24 mammal species crossing the transect are recorded, usually from mid-January to mid-March, when snow conditions are good. Considering the distance surveyed (typically 12 km) and the number of days since last snow fall, a snow track index representing the number of snow tracks/10 km/day is calculated. We used records collected between 1989 and 2019. The spatial location of each triangle was available as the center point of the triangle.

**Understory vascular plants**. Understory vegetation was surveyed within a systematic network of just over 1700 sites established on mineral-soil in forested land between 1985–1986 (as part of the 8th Finnish National Forest Inventory[40]). This network consists of clusters, which are located 16 km apart from each other in southern

Finland, and 24 and 32 km apart in northern Finland along east-west and north-south axes, respectively. All sites were resurveyed in 1995, and a subset of 443 sites were resurveyed in 2006. The spatial extent of sampling was comparable across surveys covering the whole country. In all three surveys, vascular plant species (consisting of small tree and shrub seedlings and saplings up to 50 cm height, dwarf shrubs, herbs and graminoids) were identified and the cover of each species was visually estimated on four permanent square-shaped sampling plots of 2 m$^2$, located 5 m apart from each other within each site. The average of species cover across these four sampling plots is used as an estimate of species abundance at each site[41–43]. The data are curated by the Natural Resources Institute Finland (Luke). The spatial location of each vegetation survey site is represented by the central point of the line along which the four sampling plots are located.

**Phytoplankton.** The national Finnish phytoplankton monitoring database maintained by the Finnish Environment Institute (SYKE; Open data portal http://www.syke.fi/en-US/Open_information) comprises nationwide phytoplankton community data of lake surface water samples. We selected data collected in the late summer months (samples taken during early July to late August), to reflect the peak productivity season of lake phytoplankton communities. For this study we used phytoplankton data collected between the years 1980 and 2017. All phytoplankton samples were preserved with acid Lugol's solution and analysed using the standard Utermöhl technique[44]. The spatial location of these surveys is given by the coordinates of the sampling points.

## Data filtering
We removed all records that were not identified to the species level, and records of species other than the focal species group in each dataset, e.g. bird observations recorded in the game triangle scheme (mammal dataset), or mammal observations recorded in the bird transect scheme. All records were converted from abundance values to occurrences (see below under *Joint Species Distribution Modelling*).

## Protection status
We obtained spatial data on existing designated PAs, including private and state-owned areas, as of March 2021 (World Database on Protected Areas[45]). To classify the survey sites within protected versus unprotected areas, we defined a circular buffer around each survey site and intersected that buffer with the PA layer. If the buffer intersected with a PA, the survey site was considered *protected*, otherwise *unprotected*. This approach builds on two assumptions. First, the use of a circular buffer assumes that any protected area close enough to the survey sites will influence the occurrences recorded in those locations. For the bird data, we set a buffer with a radius of 500 m around each transect line. For the mammal data, we set a buffer centered at the center of the triangle and extending 500 m beyond its vertices. For the plant data, we set a buffer of 100 m radius from the center of the sampling site. We opted for a narrower buffer for the plant data compared to the bird and mammal data because plants are sessile, and thus 100 m is deemed a reasonable distance balancing spatial effects with the precision of the available coordinates for the plant survey sites. For the phytoplankton, rather than setting a buffer, we classified a site as protected if part or all of the lake was protected. Second, we follow e.g. ref. 15 in assuming that the effect of protection is independent of the exact year in which the PA was formally established. Our results are robust to both of these assumptions as variation in the year of PA establishment (Supplementary Figs. 4, 5) and the coverage of the buffer by PAs (Supplementary Figs. 8–11 and Supplementary Table 2) yielded qualitatively similar patterns (or even reinforced them in the case of timing of PA establishment) as those of the main results.

## Environmental variables
To match protected survey sites with otherwise similar but unprotected sites, we extracted a set of environmental variables around each survey location; specifically, we used the Corine Land Cover (hereafter CLC) vector dataset for the year 2000[46]. From the CLC habitat classes, we considered eight classes as influencing site suitability across the different taxa in our study: artificial surfaces (CLC class 1), wetlands (CLC class 4), waterbodies, including inland and marine waters (class 5), arable land (class 2.1), shrub and/or herbaceous vegetation (class 3.2), broad leaved forest (3.1.1), coniferous forest (3.1.2), and mixed forest (3.1.3). The cover of each of these land use classes was determined within a 1 km radius around each survey location. For phytoplankton, the buffer extended from the lake shore to 1 km inland, thus excluding the lake surface. We transformed the cover values to proportions over the total buffer area, which were then used to match protected sites with unprotected sites with similar environmental features (see section *Matching protected and unprotected sites*). Because phytoplankton are affected by lake properties[47,48], we obtained four additional relevant covariates for the phytoplankton survey sites, namely the size of the lake (from lake contour shapefile), water color, total phosphorus concentration, and total nitrogen to total phosphorus ratio (the latter three derived from the average of the measures taken when sampling for phytoplankton during field data collection).

Furthermore, each survey location was assigned to one of the three main vegetation zones of Finland[49]: North Boreal, Middle Boreal, and South Boreal (which, for this study, also included the Hemi-boreal zone, as the latter is too spatially restricted to be considered separately; Fig. 1). Including the vegetation zone in the matching variables allowed us to account both for the biogeographical effect on species occurrence and for the spatial bias towards larger protected areas in the north of the country where land productivity and thus also human pressure is lower (see e.g. refs. 27,50; see Supplementary Figs. 12, 13 for sensitivity of results when the North Boreal zone was excluded). Overall, the above variables are meant to capture, at least to a large extent, the effect of confounders on the treatment (i.e., the placement of PAs, largely through the inclusion of the vegetation zones) and on the outcome of interest (i.e., species occurrence, largely through habitat variables; but see the next section regarding potential unobserved confounder effects).

## Matching protected and unprotected sites
To reach a balanced study design minimizing potential confounding effects stemming from landscape differences in protected and unprotected sites[24], we employed a matching approach. Matching is typically used to identify comparable pairs of treated and control units, which will only differ in respect to the focal treatment[20]. For conservation impact-evaluation studies, the treatment is often the protection status of a site[12,14]. While we aimed to capture as best as possible the potential effect of confounders on both the treatment and outcome of interest, we cannot exclude that some potential unobserved confounder variables exist, and this caveat should thus be borne in mind when interpreting the results. Capturing all potential confounders in such a complex and species-rich system is challenging[51,52]. For example, the variables used here for matching largely capture landscape and ecoregion (vegetation zone) level environmental parameters, but may neglect processes acting at the local level (e.g. land tenure), which might affect the treatment and the outcome. On the other hand, at the local scale we considered the proportion of artificial surfaces, which at least partly captures variation in human population density or influence.

To achieve comparable site pairs, we matched (using the R package *MatchIt*[53]) each protected site with the most similar unprotected site based on the eight environmental variables detailed above (12 for phytoplankton data). Furthermore, protected and unprotected

sites were selected within the same vegetation zone, and with a similar mean year of the site survey history. We also included sampling effort as an additional matching covariate for birds and mammals, as survey effort was not constant among sites. We chose an approach based on one-to-one nearest-neighbor covariate matching without replacement, using a caliper value of 0.25 standard deviation of the propensity scores[13]. The caliper value (range 0 to 1) restricts the matching of treated and control units within a certain range of the values of the covariates or the propensity scores[20]. Essentially, a very low caliper value would impose very strict (almost exact) matching conditions between covariate values of the treated and control units, potentially resulting in a high number of treated units not being matched to a correspondent control unit, because the criteria are too stringent.

To arrive at the final matching scheme, we tested the performance of two commonly applied distance based matching methods: the Mahalanobis distance metric and the propensity score matching[12–14,54]. For both methods, we assessed performance based on the absolute standardized mean difference between the covariate values from the treatment and control groups[13] (Supplementary Figs. 14–21). For birds, mammals, and phytoplankton, the propensity score matching gave better results than the Mahalanobis distance method. In other words, this method was able to efficiently identify more similar pairs of protected and unprotected sites across most of the criteria considered, while retaining the same number of matched pairs. Conversely, for plants the Mahalanobis distance method performed better. The majority of protected sites were matched with a comparable unprotected site: 86% of 360 protected sites for birds, 95% of 937 protected sites for mammals, 90% of 366 for phytoplankton, and 90% of 62 for plants. After matching, the data consisted of 311 unique protected sites for birds, 888 for mammals, 329 for phytoplankton, and 56 for plants, and an equal number of unique unprotected sites for each taxonomic group (see Fig. 1 for the distribution of matched protected and unprotected sites for each taxonomic group).

## Joint species distribution modelling

We analysed occurrence data for each of the four taxonomic groups separately, due to differences in the sampling methods and data structure between groups. We followed[35] in excluding species with <10 occurrence records across the whole study period. For each group we fitted a latent-variable joint species distribution model using the Hierarchical Modeling of Species Communities (HMSC) framework[23]. HMSC is a multivariate Bayesian linear mixed effects modeling approach which allowed us to account for the correlation among species responses, as well as for spatial and temporal autocorrelation in the data[23,55]. The main advantage of this approach compared to, for example, a frequentist generalized linear mixed effects model is that, by leveraging the information on species co-occurrence, it allows for robust modeling also for the more rare species[23]. However, conceptually, the results yielded using the approach employed here can be interpreted in a similar way as those presented by a recent similar study using frequentist modeling[13]. That is, our estimated values for the slope are conceptually comparable to those of the slope of the effect reported in[13], as both estimate each species response to the explanatory variables included in the models. Models were fitted using the Hmsc R package[55] in R[56] (version 4.0.3), assuming a binomial error distribution and probit link function suitable to analyse occurrence data (modeling occurrence probability), and adopting the default prior distribution[23], closely following the procedure applied by a recent similar study[35]. As fixed effects in each model we included the protection status (categorical variable: protected vs unprotected), year (as continuous variable; rescaled so that year one represents the earliest survey year for each dataset), and the interaction between protection status and year. Our main interest is in the interaction term, which quantifies whether

occurrences within PAs followed a different trend compared to trends in unprotected areas. In addition, we included the vegetation zone (three classes) to account for unexplained biogeographic factors. To account for the variation in sampling effort among survey sites, we also included survey effort (log transformed) as an additional covariate for birds and mammals, and lake size (log transformed) in the phytoplankton model. To account for spatial and temporal autocorrelation, we included a spatial random effect for site and a temporal random effect for year. The temporal random effect was omitted for plants because these data consist of only three survey years (see above).

We set four Markov chain Monte Carlo (MCMC) chains, with a minimum thinning of 1000 and retrieved 250 samples per chain to achieve 1000 posterior samples per model, which is deemed adequate[23]. Model convergence was assessed by investigating the potential scale reduction factor (PSRF)[57]. PSRF measures how well the MCMC chains have mixed, with values below 1.1 indicating adequate mixing and satisfactory model convergence leading to reliable posterior estimates[23]. Model fit was assessed by quantifying model accuracy using the root mean square error (RMSE), which measures how close the best predictions are to the observed data. As a measure of discrimination, i.e., how well the model correctly classifies a sampling unit as either occupied or empty, we used the area under the curve (AUC)[58]; and Tjur R2[59]. All models achieved satisfactory convergence (PSRF << 1.1) and good fit (e.g., AUC >> 0.75; Supplementary Table 3).

We evaluated the effects of protection aggregated at the level of higher taxonomic groups (e.g. the number of species with statistically supported positive effect of protection), as well as for individual species. We classified species responses to protection following the same approach employed in a recent study[35] also using the HMSC framework. Specifically, we identified the species that did not respond to protection if the posterior distribution of the corresponding slope beta parameter estimates included zero with a probability of >10% (i.e., having <90% posterior probability for the response). The non-zero responses were then classified as either positive or negative based on the sign of the beta parameter for the protected area * year interaction term. For the individual species, we present the full distribution of how protection of the site is estimated to affect the occupancy of the species. In addition, we split out the effects for species with different overall occurrence trends within and outside of PAs, i.e., for species increasing vs decreasing nationally regardless of the protection status of the site.

## Species-level trait data and analyses

We collated species-level traits referring to ecological requirements and/or life history (size), thermal niche (species temperature index), and conservation status. Size was obtained from the following sources: birds[60], mammals[61], plants (plant height[62]), phytoplankton (cell volume[48]). Species temperature index (STI), which is a measure of the average long-term temperature experienced by a species across its range[63], was extracted for birds, mammals and plants following[64]. STI was not extracted for phytoplankton due to lack of available range maps for this group. The IUCN conservation status was obtained from[65] and was categorized into two classes: threatened (including species with IUCN status from Critically Endangered to Near Threatened), not threatened (IUCN Least Concern). This information was only available for birds and mammals.

The correlation between species level response to PAs (as derived from the main model detailed above) and species traits (namely body size, species thermal niche and IUCN conservation status) was assessed using linear regressions (see results in Supplementary Table 1). The response of the model included the species-specific effect of protection as estimated by the model presented in the main manuscript, Fig. 2, with values ranging from negative to positive. To account for taxon-specific differences, we fitted a separate model for each of the

four taxonomic groups, assuming a Gaussian error distribution and an identity link. Prior to model fitting we log transformed body size and ensured that there was no collinearity among the predictors. Models were validated by inspecting the residuals against departures from a normal distribution[66].

In addition, we tested if there was a phylogenetic signal in species responses by calculating the value of K and λ[67] for each taxonomic group separately. There was no measurable phylogenetic signal in the effect of PAs within the taxonomic group (Supplementary Figs. 22–25). Phylogeny for each group was obtained from: birds[68], mammals[69], plants[70,71] and phytoplankton (following the approach of ref. [48]).

## Reporting summary
Further information on research design is available in the Nature Portfolio Reporting Summary linked to this article.

## Data availability
All the wildlife occurrence data used for the analyses are available in an online repository at Figshare[72]. The raw wildlife observation data, original data before filtering, are available from different sources and with different conditions. Bird data are available through the Finnish Biodiversity Information Facility, FinBIF (https://laji.fi/en/). Mammal and plant data are available upon request from the Natural Resources Institute (LUKE—requests should be sent to kirjaamo@luke.fi). Phytoplankton data are available through the open access data service of Finnish Environment Institute (SYKE). Spatial data from the WDPA World Database on Protected Areas (accession time: January 2021) are freely available at www.protectedplanet.net. The land-use variables (accessed on January 2021) are freely available from ref. [46] (https://land.copernicus.eu/pan-european/corine-land-cover/clc-2000).

## Code availability
The codes used for analyses in this study are available in an online repository at Figshare[72].

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

## Acknowledgements

We are grateful to all the volunteers and researchers who have collected and curated the data over several decades. This project received funding from the European Commission through the Horizon 2020 Marie Skłodowska-Curie Actions individual fellowships (Grant no. 101027534) to A.S., from the Jane and Aatos Erkko Foundation to A.S., O.O., T.R., J.V., L.A., M.Saa., from the European Research Council (ERC) under the European Union's Horizon 2020 research and innovation programme (grant agreement No 856506; ERC-synergy project LIFEPLAN) to O.O. and T.R., from the Academy of Finland, grant no. 309581 to O.O., grant no. 322266 to T.R., grant no. 330739 to M.H., grant no. 317255 to J.V., grant no. 340280 to L.A., grant no. 312650 (Project BlueAdapt) to B.W., from the Research Council of Norway through its Centres of Excellence Funding Scheme (223257) to O.O., from the Finnish Cultural Foundation to E.K., from the Helsinki Life Science Institute to M.Saa. The present research was carried out within the framework of the activities of the Spanish Government through the "Maria de Maeztu Centre of Excellence" accreditation to IMEDEA (CSIC-UIB) (CEX2021-001198). Open Access for this article was partly funded by Helsinki University Library.

## Author contributions

A.S. conceived the study, in discussion with B.W., L.H.A., E.K., M.H., O.O., J.V., T.R., and M.Saa.; A.S. carried out the statistical analyses, with the support of O.O., L.H.A., J.V., A.Li. and B.W.; A.S. wrote the first draft. A.S., B.W., L.H.A., E.K., M.H., A.Le, A.Li, M.Sal., T.T., P.M., K.V., O.O., J.V., T.R., M.Saa. interpreted the results, and commented and revised the manuscript.

## Competing interests

The authors declare no competing interests.
