## [Peer Review File · Nature Communications]

Reviewers' Comments:

Reviewer #1:

Remarks to the Author:

Santangeli et al review

This is a fascinating paper on an extremely important topic presenting a valuable analysis of an exceptionally rich dataset. I strongly recommend it is published.

I have some mostly pretty small comments.

1) The analysis is (by necessity) complex with some inevitably somewhat arbitrary modeling choices. To advance science, I think it is essential that reproducible code (which takes readers from data through to these analytical outputs) should be made available. On reflection, it seems like the code and data are available in a FigShare (accessed via a SI). Therefore I am not clear why the main text said 'code available on request'? In summary I don't think there is a problem (I haven't checked the code runs but did have a quick look).

2) The section explaining matching seems mostly sensible except, little or no attention is given to the question of unobserved confounders. There will inevitably be reasons some sites were protected and others were not. The selection of matching covariates is ultimately meant to reflect confounders (anything which affects 'selection to the treatment' and the outcome of interest). Perhaps the landscape variables used captures this perfectly, but quite possibly not. I think this deserves more attention (eg see Rasolofoson et al 2022 and Jones et al 2022 which, in a follow-up discussion to Schliecher et al 2020 which is cited, discuss this point).

3) There is something I am not fully understanding in the analysis. They are focused on the interaction between date and PA status (to explore if species followed different trajectories inside and outside PAs). I understand this and it makes perfect sense and is beautifully presented in Fig 4 (which is an excellent piece of data visualization). However I am not an expert in SDMs and I am afraid I don't fully get how this model is fitted as seems to be inside a SDM. I am not necessarily saying this is wrong in anyway, but I do encourage the editor to ensure they have had someone who fully understands this aspect of the analysis look at the paper. You cite Wauchope et al 2020 in a few places (very appropriately). I wonder if being more explicit about the differences of approach and reasons for using the SDM rather than just the sort of linear models suggested in Wauchope et al might help as that paper tries to explain impact evaluation from species time-series data (note conflict as I am an author of that paper-no pressure to do this of course if it isn't helpful).

4) The summary could easily be improved. It feels odd to start with a statement that PAs are vital to counter biodiversity loss when the whole paper is questioning the extent to which PAs do actually do anything for biodiversity. Similarly the statement 'our results indicate that PAs alone will not.....' seems to somewhat miss the point. A major conclusion is presumably that many PAs are not effectively managed and are therefore not slowing biodiversity loss even within their borders. This seems to be missed by the way this is written. Finally, given that of the 638 species most are phytoplankton, I wonder if this statistic could be better communicated.

5) I am unclear why we need Fig 2 and Fig 3? (Fig 3 seems so much richer). However they can't be presenting the same data (as the number of species for each taxa showing 'no effect' seems remarkably fewer in Fig 3 so I assume I am missing something important).

Congratulations on a fascinating paper. I hope to see it published soon. I will certainly refer to it often.

Julia P G Jones

References (note I am an author on 1 of these, I am certainly not suggesting you need to cite them but they help illustrate points I am making above and make be helpful).

Rasolofoson et al 2022 <https://conbio.onlinelibrary.wiley.com/doi/full/10.1111/cobi.14006>

Jones et al 2022 <https://conbio.onlinelibrary.wiley.com/doi/full/10.1111/cobi.14007>

Reviewer #2:

Remarks to the Author:

The piece entitled "Mixed effects of protected areas on terrestrial and freshwater biodiversity" brings an interesting overview of the impact of protected areas (hereafter PAs) on the occurrence of different biological groups in Finland. This topic is worth investigating and can contribute to future conservationist interventions and actions interested in optimizing biological outcomes.

Studies focused on biological entities are still rare, even more considering a study at this scale and interest in various biological groups presented here. Even considering this worth investigation and the study bring interesting results, I have concerns about the counterfactual estimates built for answering the question of this study. To the best of my knowledge, the counterfactuals of this study are weak when considering different factors that influence the outcome of interest and the establishment of the intervention. Considering that the method to evaluate impact here is a post hoc approach, such details in estimating counterfactuals are fundamental to rely on the results.

Line 54. Although I agree with this statement, it is also important to say that counterfactual reasoning can improve our understanding of conservation and that this approach is increasingly common nowadays.

Lines 58-59. This last statement could be read as a combination of three factors instead of two, those being: (1) counterfactual reasoning, (2) multiple taxa investigation, (3) same spatial resolution.

Line 64. Environmental and biogeographic features. Did you use some socioeconomic features? If not, how might this affect your conclusions? Please check my last topic in this review.

Results and discussion

It might be interesting to evaluate how overall PAs sizes or other features related to protection level can affect the protection statement of these species. This comes to my thoughts looking at your Figure 1, where Finland looks to have uneven and diverse distribution and sizes of PAs. Also, results can be shown per vegetation zones and explored in this sense. We have many big pictures in literature due to counterfactual approaches. However, more focused and specific contexts still need to be added. Your data and design look relevant for such exploration.

Also, it could be interesting to show the challenges new conservation targets might face to improve the conservation of different species.

Line 87. I suggest adding a percentage at the beginning of the sentence here to improve reading and understanding.

Line 109. Do any specific characteristics explain why some species can respond positively and negatively? You could explore that to suggest best practices for species conservation when those show a strong negative impact of PAs.

Line 118-119. The statement of high statistical support needs to be revised since frequentist statistical inference does not support such a qualitative interpretation.

Lines 129-131. This is one of the most interesting results your study brings. Is there any hypothesis explaining this?

Line 134. There are some typos and edition problems all over the manuscript. This may be the result of the review version, but I suggest you look at the text edition overall.

Lines 126, 138, 140. You called those lines red and orange in different lines. I suggest you keep the same nomenclature, which, in my opinion, red fits best.

Lines 166-167. This statement appears first here during my reading; however, it should be presented before. I suggest that because this new information entirely changes the interpretation of results and understanding of how PAs impact species. Also, you could explore variations on slopes between periods before and after the PAs, this could show how PAs might affect the species and consider the periods before their establishment since your data contemplate a broad period, and this feature might be interesting to evaluate impacts.

Line 183. This statement is repeated during your text. I suggest changing this to include a few sentences in the introduction and a paragraph in the results and discussion section.

Lines 270-274. This assumption contrasts with the results presented between lines 165-187. Also, your sensitivity analyses only include years when PAs were established. However, my concern about this is when comparing years before and after protection and only for periods after protection. I do not think a sensitivity analysis, only considering the year when PAs were established, is sufficient to show the robustness of your assumptions. Different species and groups might have different answers to the time they are protected, and the period before protection might influence the response of these groups differently. Thus, a more profound investigation of the robustness of your approach should focus on time-series for periods after protection and not only for the year of PAs' establishment.

Lines 276-299. I have some concerns about using only environmental data for matching purposes. Usually, counterfactual reasoning seeks to estimate alternative states for your causal factor of interest; here the PAs. Thus, covariates to find matching groups should rely on causal theories, which have some differences in how they approach counterfactual reasoning. However, in general, those different theories have a common point of view, where counterfactuals should be based on a set of characteristics that influence both (i) your outcome of interest and (ii) the likelihood of establishing your intervention of interest at the specific areas they were established. In this sense, environmental variables are clearly related to the outcome and can be to the intervention; however, many other factors might influence the distribution and establishment of PAs and how protection can achieve their goals. Considering this, using only environmental variables can result in bias when estimating impacts.

Please check this paper for more about this concern: Coetzee BW, Gaston KJ. An appeal for more rigorous use of counterfactual thinking in biological conservation.

Reviewer #3:

Remarks to the Author:

Comments to the authors of 'Mixed effects of protected areas on terrestrial and freshwater biodiversity' (manuscript #NCOMMS-23-04671).

In this study, the authors evaluate the effectiveness of protected areas (PAs) in promoting the occurrence (i.e. rates of presences and absences) of 638 species in the boreal ecosystems of Finland through time (from the 70s or the 80s depending on the group) and across four clades (i.e. mammals, birds, vascular plants and freshwater phytoplankton). Based on extensive monitoring surveys that document changes in species occurrences through time, the authors use the now established Hierarchical Modelling of Species Communities (HMSC) framework to model the effect of conservation status (i.e. protected vs. unprotected) and time (i.e. year) on the distribution of occurrence probability at the species-level. The HMSC framework accounts for differences in sampling effort between clades as well as spatio-temporal autocorrelation. Survey sites located close enough to an official PA (according to a spatial buffer) are assumed to be under its influence. Sites classified as unprotected were not randomly sampled through the study area. Indeed, the authors chose unprotected sites displaying similar landscapes/environmental conditions as the PAs. This is a nice and important feature of the study as the trends modelled are thus independent of species-level differences in niches/ environmental preferences.

The authors find that most of the sites likely influenced by PAs display no detectable impact on species occurrences compared to sites not influenced by PAs, with some slight differences across clades: 90% and 92% of the phytoplankton and vascular plant species, respectively, show no detectable temporal trend in occurrences, whereas mammals and birds show higher proportions of detectable effects (78% and 69%, respectively). When the effects of PAs establishment are statistically supported, they are either positive or negative with no clear advantage towards positive effects. The authors also show that the presences of birds and phytoplankton species are promoted once PAs are established, thus supporting the view that PAs are not ineffective neither. Yet, this was not the case for mammals and a large variety of responses to PAs establishment were observed within clades. Consequently, the authors conclude that PAs can have a positive impact on the occurrence of species/groups but that there are not sufficient to protect entire communities. This study contributes to the ongoing and critical debate on the effectiveness of PAs as useful units for biodiversity management and conservation, and show how species-specific responses must be accounted for to achieve efficient community-level conservation.

I find the present submission to be of fine quality overall. The text is very pleasant to read since the language is of prime quality, the study is very well motivated (i.e. the introduction is excellent!) and the graphs are clear and convey the main results effectively. The authors made the effort to evaluate the sensitivity of their main findings to some of the key choices made in the numerical framework (i.e. year of PAs creation, spatial buffer around the PAs, spatial variability in PAs size, method used to match PAs with unprotected areas) and all this information is summarized in a comprehensive suite of Supplementary Materials. I do not see major flaws in the data analyses and their interpretation. The work carried out by the authors does support their conclusions in my opinion. The methodology is well explained and exhaustive enough to be reproduced (provided the authors make their data and R codes publicly available).

I understand that submissions to Nature Communications must remain quite short. However, I think the Discussion of the paper needs to include additional elements to better put the present results into perspective (see comments #1 to #4 below). I find that the last 10-15 lines or so of the main text are way less interesting/relevant and could be replaced by more insightful elements centered around the points I am highlighting below. To do so, the authors may need to run some additional numerical analyses but I am confident they will be able to manage these. To conclude, provided the authors address my comments below, I would recommend this study for publication.

1. PAs effectiveness for conserving the abundance/biomass of the species/clades ?

At line 254, the authors mention that all quantitative abundance values were converted to presence-absence to run the JSDMs that are part of the HMSC framework. Is this because species counts are not comparable across the four clades or is because the JSDMs cannot handle abundance data? Please clarify. Also please document the number (in relative contribution for instance) of records/taxa that were discarded because they were not identified at the species-level. I think readers should know how representative the final dataset is compared to the full survey data. How many taxa/sub-groups are lost because of this filter?

I am surprised that the authors do not mention the lack of abundance data as a critical limitation/assumption of their study. Evidently, species can be present in varying levels abundances. For ecosystem management and for evaluating ecosystem functioning, species abundance/biomass should be more informative than occurrence. PAs could be misclassified as effective if they only manage to retain species in very low quantities (i.e. the populations of the species are barely surviving). This is something that should definitely be mentioned and discussed in the main text (see Coetzee 2016, <https://doi.org/10.1007/s10531-016-1235-2>; Johnston et al., 2013 <https://doi.org/10.1038/nclimate2035>; Gray et al., 2016 <https://doi.org/10.1038/ncomms12306>), especially for a journal with such a wide audience as Nat. Comms.

I understand how the different surveys can be difficult to integrate as their methods can be hardly inter-comparable. Yet, I wonder if parts of the present dataset could not be used to assess the impact of PAs on abundance rather than occurrences (i.e. not necessarily through JSDMs). Within clades, have the authors tried to examine time series of species abundances? If possible, I would strongly encourage the authors to try to integrate this in their manuscript, at least to discuss this major caveat.

2. PAs effectiveness for conserving species diversity (alpha and beta, phylogenetic) and functions (functional diversity indices) ?

Similarly, I wonder if the authors cannot integrate diversity indices in their study to evaluate the performance of the PAs for conserving biodiversity rather than species-levels occurrences. Would not these indicators be better suited to assess community-level patterns? Have the authors tried to look at the spatio-temporal dynamics of indices such as Shannon-Wiener or Simpson's (alpha diversity) or Bray-Curtis distances (or Jaccard's or Sørensen's; for beta diversity)? Similarly, could the efficiency of the Finnish PA network vary across the phylogeny or the functional traits of the species? Considering that a wealth of phylogenies and functional traits compilations exist for the clades studied (except maybe for the phytoplankton), I wonder if the authors could not integrate indices of phylogenetic diversity or functional diversity in their analyses? Or maybe they are considering integrating those in follow-up studies? If the goal of the authors is indeed to examine the effects of PAs on Biodiversity (i.e. the title of the study), I feel like they should go beyond occurrences and integrate more dimensions of Biodiversity.

3. How can functional traits help explain differences in PAs effectiveness across clades ? Are there any trends within clades (e.g. Order-levels? Family-level? Genus-level?) ?

Along the same lines as above, I strongly encourage the authors to better discuss the potential effects of species' traits (or clade-level traits) on the observed differences in trends? The authors find that the rates of occurrence increase post-PA establishment for birds and phytoplankton, but not for vascular plants and mammals. This is quite interesting as birds and plankton could be viewed as groups that are less subject to dispersal limitations than vascular plants and mammals. Overall, the four clades display major differences in life histories, life cycle length, physiologies, thermoregulation (ectothermy vs. endothermy), dispersal capabilities etc. How can those explain differences between clades? What about intra-clade variability in traits? What biases are introduced when comparing phytoplankton that passively live in interconnected lakes (how well are those lakes connected in Finland?) and birds that can fly miles to nest in the habitat that they find suitable? Furthermore, could the authors did try to interpret/explain the high level of within clade variability (Figures 3 & 4) through species specific traits? Among the positively/negatively impacted taxa, are there specific sets of traits that could explain the positive/negative influence of the surrounding PA? Are there any intra-clade patterns from a taxonomic point of view (family-level, order-level etc.)?

The authors perform JSDMs but do not seem to make use of all the information those can give. Following Pollock et al. (2014), « JSDMs [...] allow decomposition of species co-occurrence patterns into components describing shared environmental responses and residual patterns of co-occurrence ». Therefore, I was wondering if the authors could not identify sets of species (within clades) that display co-occurrence patterns driven by shared environmental responses, and test whether those sets display similar temporal dynamics in PAs (e.g. same responses in Figures 3 & 4).

4. Is there an effect of PA category on species occurrences? Can the features and enforcement levels of PAs help explain differences between and within clades ?

Another major caveat of the study that the authors should better discuss in the main text is the fact that not all PAs are equal. As the authors must be aware of, the IUCN defines no less than seven categories of PAs, from « strict reserves » to « areas with sustainable use of resources ». First, I encourage the authors to mention the types of PAs that are present in their study area (i.e. the relative contribution of each IUCN PA category; a supplementary plot could be shown; or color the points of Figure 1 as a function of IUCN status? Which makes me think that there could be an interaction between space and the status of the PA)? Second, could these categories be encoded as hard/random effects in the HMSC? I expect the level of protection/ enforcement to have a major impact on the efficiency of the PAs. Therefore, could the positive trends be driven by PAs characterized by more stringent levels of conservation? Would it also make sense to make the spatial buffer around the PAs vary also as a function of conservation status (i.e. the level of PA influence could increase with the level of conservation)? Overall, I strongly suggest to better discuss this in the manuscript (unless all the PAs of the study area have the same status?).

5. Selection of environmental covariates (more minor comment).

The authors rely on the CLC data to classify survey sites according to their environmental similarity with the PAs. Four additional parameters (lake size, lake water color, phosphorus concentration and nitrogen to phosphorus ratio) were included for freshwater phytoplankton. I was wondering why the authors did not choose to include lake surface temperature for phytoplankton (and vascular plants) as well? There is now widespread evidence that temperature controls the physiology, distribution and production of freshwater phytoplankton species. I find that this is a key factor for measuring the similarity of survey sites for phytoplankton and plants, as not all species can occur between areas that vary sometimes by even $< 5^{\circ}\text{C}$.

Code availability: to abide the FAIR principles for data sharing and stewardship, I would ask the authors to make their data and R codes already available on an online repository such as Zenodo or GitHub.

I hope the authors will find my comments useful and that they will use them to improve their interesting study.

Yours faithfully,

Reviewer #4:
Remarks to the Author:

Mixed effects of protected areas on terrestrial and freshwater biodiversity

Submitted to: *Nature Communications*

Authors: Santangeli et al.

General comments:

Thank you for the opportunity to review this manuscript. Although I do appreciate the effort and analysis put forth by the authors, I do not think the contribution warrants publication in the journal *Nature Communications*. I do think, however, that the work is publishable in a different outlet.

I provide comments below that explains my decision, but I think the “impact” and contribution of this work was best summarized by the authors in the last sentence of their Abstract that states “Better monitoring and robust assessment of their effectiveness are fundamental to leverage the benefit of protected areas and to bend the curve of biodiversity loss.” I think this statement is an accurate reflection of the analysis and results, but unfortunately, if the main take-home message is that we need to collect more (“bettter”) data, then it is my opinion that the contribution and impact of this work is not suitable for this outlet. The main results of whether or not a species showed a “positive” or “negative” response to Protected Area (PA) was not very compelling for the following reasons:

- I found the categorization of “positive”, “negative”, and “no effect” difficult to interpret ecologically. For example, a PA had a negative effect if there were slower (less positive increase) rates of increase in occurrence probability compared to outside of PAs. Without species level information and a better discussion of effect sizes, this “negative” effect is really difficult to assess or even to know if it is in fact “negative” from an ecological perspective. It may be that a less rapid increase in occurrence probability over time in a PA is a “positive” ecological effect given the species and effect size. The authors acknowledge this starting line 155, but I find the this issue is severe enough that it really limits the usefulness of the assessment. Also, the overall effectiveness of PAs for most species was “no detectable effect” - again, this may be a “positive” effect in some cases. For common species showing no negative trend outside a PA, perhaps having no detectable trend in the PA is a good (positive) outcome? These “no detectable effect” species are the majority, but there is little additional investigation or discussion about them.
- Related to the above point - there is no discussion of what types of species within the broad taxonomic groupings may or may not be responding in some way to PAs – other than designating species as “threatened” or “non-threatened”, were there certain species traits that corresponded to type of PA response (assuming you can interpret those responses in a meaningful way)? I think some treatment of species-level characteristics is needed to make these types of analyses useful for practitioners. For example, the authors make the argument that there is an urgent need for this type of analysis, but

without some investigation at the species-level, I question its utility to actually inform the management, use, etc. of PAs. E.g., just knowing the % “birds” that showed a “positive” response to PA does not seem that useful to guide management and climate adaptation strategies.

- Generally a better treatment, illustration, and discussion of effect sizes, if they are ecologically relevant (not just statistically significant), is needed. The authors provide the protected area effect in Figure 3, but the logit-scale interaction term is not easily interpreted for ecological relevancy.

Other comments:

- Abstract: Line 31 (and throughout) – what are the effect sizes you are talking about here and are they ecologically significant/meaningful. “Slower rates of decline” cannot be interpreted in an ecologically meaningful way.
- Introduction: Line 37 – “ecological impacts” on what? Are you just talking generally about global-scale impacts of humans or specific impacts on certain systems?
- Line 50: I would rephrase “. . . robust PA impact evaluation is problematic” to “. . . assessment of PAs is problematic
- Line 50: Some brief discussion of how PAs are evaluated, if not using a counterfactual approach, is warranted.
- Line 54: I would avoid using terms like “even worse” (worse than what?), “easily” (line 56), “urgent” (line 58), “striking” (line 85), etc. The use of these terms are vague, not quantifiable, and sometimes give the impression of trying to increase the visibility and contribution of the work. For example, the sentence on line 85 that states the lack of protection effect was particularly *striking* for plants and phytoplankton – I did not find these results particularly striking given the ecology of these organisms and the reliance on occurrence data.
- Line 56: This analysis is focused on PA effectiveness in terms of preserving biodiversity, but they serve other functions (e.g., reducing carbon fluxes to the atmosphere) that are considered during the development of PAs, but this is not touched on in this paper. I think a sentence of two of the broader ecological importance of PAs is warranted.
- Results and Discussion: Line 82: I think you need to start the results with a discussion of the what the trends were in PAs and outside PAs before you jump into if species were “protected”. Again, I find the lack of discussion of effect sizes makes it difficult to assess importance of the work. What were ranges in the probability of occurrence over time for different species?
- Figure 3: caption and figure – it states “. . . where darker colors indicate higher statistical support for the effect.” I am left to assume that this is the posterior probability of a positive or negative effect, but that should be stated, if so. Alternatively, to improve

readability of this figure, just shade the ones with Prob>90% to align with your aforementioned cutoff. The use of a probability cutoff above and a gradient here does not help readability (although I appreciate not relying on a cutoff of artificial significance generally).

- Figure 4: What about no effect in or out of PAs, which are the majority of the results. Again, I think it is potentially inaccurate to suggest that “no effect” is not potential “protection” for some species – especially if the trends in and out of the PA are relatively flat (it may not be protecting better than sites outside the PA, but still may be protective in the broader sense of preserving habitat and having longer term benefits by being a PA).
- It seems the authors have the expectation that PAs should be “protective” for all species for the PA to be considered effective. Would we really expect all species to increase in occurrence probability in a PA over time? This seems like a very high standard and one which will lead to the results presented – that PAs are perhaps not great at preserving biodiversity. I understand the authors caveat this finding, but nonetheless it is one that people will likely take away from this paper.
- Line 149 states “Generally, only the cases where PAs actually mitigate or reverse declines can be seen as unequivocally positive conservation outcomes. ” I think this statement oversteps a bit – what can be viewed as a positive conservation outcome is dependent on the goals and objectives of the PA and might be species specific. I think this statement simplifies the equation too much,
- Lines 171 – 177: Again, the authors acknowledge that the main contribution of their work is to suggest that we need “rigorous assessment” – a finding that I don’t think warrants publication in *Nature Communications*.
- Line 181 states “. . . for improving conservation practice and avert the biodiversity crisis” – the crisis is here my friends, there is no averting it. We can try to minimize its magnitude, but it will not be averted.
- Statistical analysis: I appreciate the thoughtfulness of the treatment of the data, the analysis, and the assessments of results to assumptions and model structure and data decisions (i.e., the sensitivity analyses).

Minor comments:

- Supplement tables 1 and 2: I know this is a regional preference, but it might be better presented to use decimal points throughout instead of decimal commas as in Table S1.

Response letter.

Authors' replies in italic font

Reviewer #1 (Remarks to the Author):

Santangeli et al review

This is a fascinating paper on an extremely important topic presenting a valuable analysis of an exceptionally rich dataset. I strongly recommend it is published.

I have some mostly pretty small comments.

Thank you for your positive and encouraging comments on our work.

1) The analysis is (by necessity) complex with some inevitably somewhat arbitrary modeling choices. To advance science, I think it is essential that reproducible code (which takes readers from data through to these analytical outputs) should be made available. On reflection, it seems like the code and data are available in a FigShare (accessed via a SI). Therefore I am not clear why the main text said 'code available on request'? In summary I don't think there is a problem (I haven't checked the code runs but did have a quick look).

We confirm that code and data are now openly available in Figshare repository, see link in the manuscript. We apologize for the lack of clarity and correspondence between the previous manuscript text and the reporting summary. This has now been corrected and clarified.

2) The section explaining matching seems mostly sensible except, little or no attention is given to the question of unobserved confounders. There will inevitably be reasons some sites were protected and others were not. The selection of matching covariates is ultimately meant to reflect confounders (anything which affects 'selection to the treatment' and the outcome of interest). Perhaps the landscape variables used captures this perfectly, but quite possibly not. I think this deserves more attention (eg see Rasolofoson et al 2022 and Jones et al 2022 which, in a follow-up discussion to Schliecher et al 2020 which is cited, discuss this point).

We are happy to see that the Reviewer finds our approach sensible, and we thank her for bringing up these very relevant points regarding confounders. We are fully aware of the issues of unobserved confounders affecting the treatment (protected area placement) and the outcome (species response), and we indeed followed the protocol presented by Schliecher et al. (2020) - which initiated the constructive discussion published in the two papers mentioned. With four very contrasting and species-rich taxonomic groups, we had to carefully balance robustness and practicalities. We believe that we have achieved a close-to-optimal design given the complexity of the study system, the available data and the number of species involved. Crucially, regarding the treatment effect, the main driver of protected area establishment in Finland is location across latitudes, because the land is most productive, and most expensive, in the south. Hence most of the large protected areas are placed in the north of the country (see e.g. Virkkala & Rajasärkkä 2007). We have accounted for this effect by imposing an exact match of treatment and control within each bioclimatic zone of Finland, which captures latitudinal gradients. Moreover, we also had a matching variable for artificial surfaces, which captures, at least partly, human population density. Regarding confounders on species responses, we tried our best to include as relevant

variables as possible that would apply broadly to most of the species, on one hand, and on the other we also took a taxon specific approach to refine this set of variables. This was particularly relevant for phytoplankton where, in addition to landscape variables, we also included water parameters (such as water color and Nitrogen levels) which are known to strongly affect phytoplankton communities (Weigel et al., 2023). Nevertheless, we fully share the reviewer's view on the importance of unobserved confounders, and we now added text in the methods explicitly mentioning this aspect (Lines 342-347). Please also see our reply to a similar comment by ref. 2.

*Virkkala, Raimo, and Ari Rajasärkkä. "Uneven regional distribution of protected areas in Finland: consequences for boreal forest bird populations." *Biological Conservation* 134.3 (2007): 361-371.*

*Weigel, Benjamin, et al. "Macrosystem community change in lake phytoplankton and its implications for diversity and function." *Global Ecology and Biogeography* 32.2 (2023): 295-309.*

3) There is something I am not fully understanding in the analysis. They are focused on the interaction between date and PA status (to explore if species followed different trajectories inside and outside PAs). I understand this and it makes perfect sense and is beautifully presented in Fig 4 (which is an excellent piece of data visualization). However I am not an expert in SDMs and I am afraid I don't fully get how this model is fitted as seems to be inside a SDM. I am not necessarily saying this is wrong in anyway, but I do encourage the editor to ensure they have had someone who fully understands this aspect of the analysis look at the paper. You cite Wauchope et al 2020 in a few places (very appropriately). I wonder if being more explicit about the differences of approach and reasons for using the SDM rather than just the sort of linear models suggested in Wauchope et al might help as that paper tries to explain impact evaluation from species time-series data (note conflict as I am an author of that paper-no pressure to do this of course if it isn't helpful).

Thank you for this relevant comment, which made us realize that we had not been fully explicit about why we used the HMSC approach over traditional frequentist ones, such as the GLMM used by Wauchope et al 2020. We have now added more details to the methods, highlighting the advantage of the HMSC approach (joint species distribution models) and how it compares to GLMM, especially with regards to the interpretation of the results (lines 379-385). Importantly, the outcomes of the two approaches are highly comparable. HMSC is an extension of GLMM, which: 1) accounts for confounding spatio-temporal patterns in the random part of the model; and 2) improves the inference for rare species through interspecific correlations between fixed effects. Hence, while HMSC accommodates Spatial Distribution Modeling, we applied it as a regression type of approach that allowed us to estimate species responses to protection over time adopting the same rationale as used in implementing HMSC in a recent study (<https://www.nature.com/articles/s41558-022-01381-x>).

4) The summary could easily be improved. It feels odd to start with a statement that PAs are vital to counter biodiversity loss when the whole paper is questioning the extent to which PAs do actually do anything for biodiversity. Similarly the statement 'our results indicate that PAs alone will not.....' seems to somewhat miss the point. A major conclusion is presumably that many PAs are not effectively managed and are therefore not slowing biodiversity loss even within their borders. This seems to be missed by the way this is written. Finally, given that of the 638 species most are phytoplankton, I wonder if this statistic could be better communicated.

Thank you for this suggestion. We agree that the summary needed improvements, and your comments gave us useful pointers towards this end. Regarding the observed mixed effects of PAs

on species occurrence probabilities, we believe that there are likely two processes affecting this. One, as the Reviewer points out, is the effectiveness in the management of many PAs. The other is the size and connectivity of PAs, in the central and southern part of the country especially, which does not allow the whole network to effectively prevent biodiversity loss under anthropogenic changes. The population dynamics of many of the species analyzed may depend on processes occurring at a much larger scale than the mean extent of a PA. An important reason for why we see a slowed-down rate of species loss inside PAs may be that individuals of the remaining populations aggregate in the high-quality habitats remaining inside PAs. Indeed, a recent study (targeting birds alone) showed that the coverage of the Finnish PA network plays a key role in buffering communities against climate-driven reshuffling (Lehikoinen et al., 2021). We have thus amended the summary to include the above interpretations and conclusions, which were also added to the revised discussion (lines 165-170).

Lehikoinen, P., Tiusanen, M., Santangeli, A., Rajasärkkä, A., Jaatinen, K., Valkama, J., ... & Lehikoinen, A. (2021). Increasing protected area coverage mitigates climate-driven community changes. *Biological Conservation*, 253, 108892.

5) I am unclear why we need Fig 2 and Fig 3? (Fig 3 seems so much richer). However they can't be presenting the same data (as the number of species for each taxa showing 'no effect' seems remarkably fewer in Fig 3 so I assume I am missing something important).

Thank you for pointing out this lack of clarity. We have now moved Figure 2 to the Supplementary Material, as you are right that the information given in those is partly redundant. The figures do present the same data, but in different ways: (former) Fig. 2 with responses categorized as positive, negative and no effect, and (former) Fig. 3 presenting continuous protected area effects.

Congratulations on a fascinating paper. I hope to see it published soon. I will certainly refer to it often.

Julia P G Jones

References (note I am an author on 1 of these, I am certainly not suggesting you need to cite them but they help illustrate points I am making above and make be helpful).

Rasolofoson et al 2022 <https://conbio.onlinelibrary.wiley.com/doi/full/10.1111/cobi.14006>

Jones et al 2022 <https://conbio.onlinelibrary.wiley.com/doi/full/10.1111/cobi.14007>

Thank you for providing these references and for the time and constructive comments that helped us further improve our work.

Reviewer #2 (Remarks to the Author):

The piece entitled "Mixed effects of protected areas on terrestrial and freshwater biodiversity" brings an interesting overview of the impact of protected areas (hereafter PAs) on the occurrence of different biological groups in Finland. This topic is worth investigating and can contribute to future conservationist interventions and actions interested in optimizing biological outcomes. Studies focused on biological entities are still rare, even more considering a study at this scale and interest in various biological groups presented here. Even considering this worth investigation and the study bring interesting results, I have concerns about the counterfactual estimates built for answering the question of this study. To the best of my knowledge, the counterfactuals of this study are weak when considering different factors that influence the outcome of interest and the

establishment of the intervention. Considering that the method to evaluate impact here is a post hoc approach, such details in estimating counterfactuals are fundamental to rely on the results.

Thank you for the positive comments on our work. We are aware of the need to account for unobserved confounders, which can affect both the treatment (protected area placement) and the outcome (biodiversity response). With four very contrasting and species-rich taxonomic groups, we had to carefully balance robustness and practicalities. We believe that we achieved an optimal design given the complexity of the study system and number of species involved, as based on the following considerations: Regarding the treatment effect, in Finland the main driver of protected area establishment is location (latitude), because the land is most productive and expensive in the south, and least productive and thereby cheapest in the north. Hence, most of the large protected areas are located in the North of the country (see e.g. Virkkala and Rajasärkkä 2007). We captured this effect by imposing an exact match between treatment and control pairs within the same vegetation zone of Finland. As for the confounders on the outcome of interest, we tried our best to include as relevant variables as possible, as applying broadly to most of the species. At the same time, we applied a taxon specific approach to refine this set of variables. This is illustrated by the choices made for phytoplankton, where in addition to accounting for landscape variables, we also included water parameters (such as water color and Nitrogen levels) as known to strongly affect phytoplankton communities. Nevertheless, we fully share the Reviewer's view on the importance of unobserved confounders, and have now added text in the methods explicitly mentioning this aspect (Lines 342-347 and 330-333). Please also see our reply to a similar comment by Reviewer 1.

Virkkala, Raimo, and Ari Rajasärkkä. "Uneven regional distribution of protected areas in Finland: consequences for boreal forest bird populations." Biological Conservation 134.3 (2007): 361-371.

Line 54. Although I agree with this statement, it is also important to say that counterfactual reasoning can improve our understanding of conservation and that this approach is increasingly common nowadays.

True. We have added these two relevant points to the revised text, see lines 66-68.

Lines 58-59. This last statement could be read as a combination of three factors instead of two, those being: (1) counterfactual reasoning, (2) multiple taxa investigation, (3) same spatial resolution.

We edited this sentence to include the three factors.

Line 64. Environmental and biogeographic features. Did you use some socioeconomic features? If not, how might this affect your conclusions? Please check my last topic in this review.

We fully agree that these features are important for global and continental studies, as due to large between-country variation in e.g. GDP, control of corruption and so on (see Geldmann et al., 2019). Nonetheless, socio-economic features are likely less relevant for our country-level analysis. Within Finland, environmental and biogeographic factors will characterize the placement of protected areas, which also have a strong impact on our outcome of interest (species responses), with a key aspect being latitude. Thus, such factors were thus given particular emphasis. However, we are confident that the counterfactual variables used will also adequately capture the within-country variation in some key sociodemographic features that might affect the designation of PAs. Specifically, we note that among the variables used for matching, we included the proportion of artificial surfaces, which are strongly related to human footprint and human population density

(both are higher in the southernmost bioclimatic zone compared to the northern and central one). This aspect was already implemented at the study design level as we imposed a rule that protected and unprotected sites needed to be matched within the same bioclimatic zone.

In addition, we added a sensitivity analysis where we excluded data from the northernmost bioclimatic zone, which is the most remote and least populated zone, and where the largest PAs are situated. This analysis shows results consistent with the main ones. We nevertheless fully agree with the Reviewers 1 and 2 comments in highlighting the issue of unobserved confounders, which are now explicitly mentioned in this revised version of the manuscript (lines 342-347).

*Geldmann, J., Manica, A., Burgess, N. D., Coad, L., & Balmford, A. (2019). A global-level assessment of the effectiveness of protected areas at resisting anthropogenic pressures. *Proceedings of the National Academy of Sciences*, 116(46), 23209-23215.*

Results and discussion

It might be interesting to evaluate how overall PAs sizes or other features related to protection level can affect the protection statement of these species. This comes to my thoughts looking at your Figure 1, where Finland looks to have uneven and diverse distribution and sizes of PAs. Also, results can be shown per vegetation zones and explored in this sense. We have many big pictures in literature due to counterfactual approaches. However, more focused and specific contexts still need to be added. Your data and design look relevant for such exploration.

Thank you for these valid suggestions, which gave us a chance to further test the robustness of the results to the size of PAs, as well as their IUCN category status. Accordingly, we have now run two new analyses. One corresponds to a subset of the data including only the 50% largest PAs (which are in fact very much larger than the average size of all PAs, see Supplementary Figure S2-3), and their corresponding (matched) unprotected sites. The other corresponds to a subset of the data including only the PAs with the strictest IUCN protection status (that is, IUCN PA level I to IV), and their corresponding (matched) unprotected sites. In the latter analysis we could not include plants, due to low availability of sites (only 7) with plant data within these PA categories. These results revealed that the IUCN PA category analysis qualitatively aligns with the main results, whereas the PA size analysis revealed that when the largest PAs are considered, the effect of PAs becomes more positive for mammals, plants and phytoplankton, but not for birds. We now introduce these results in the main text (lines 116-127), and added corresponding plots in the supporting material (Figures S2-7).

Regarding the comment on presenting the results by each vegetation zone separately, we interpreted this suggestion as related to the PA size. Given that the sites are matched within bioclimatic zones and that we have now included the above described analyses, we trust that the additional exploration made will adequately account for the effect of both PA size and latitudinal distribution, as the bioclimatic zones roughly follow latitudinal clines in Finland (Fig. 1).

Also, it could be interesting to show the challenges new conservation targets might face to improve the conservation of different species.

Thank you for the suggestion, we now added two sentences (lines 193-196) discussing how the results can help inform the near future conservation targets.

Line 87. I suggest adding a percentage at the beginning of the sentence here to improve reading and understanding.

Done.

Line 109. Do any specific characteristics explain why some species can respond positively and negatively? You could explore that to suggest best practices for species conservation when those show a strong negative impact of PAs.

Thank you for this important suggestion. We have now compiled species level traits that would be relevant to the response to protection, namely:

- *body size (as a proxy for ecological resource requirements; available for all taxa),*
- *IUCN status (threatened vs not-threatened, only available for birds and mammals),*
- *Species Temperature Index (STI, a measure of the thermal niche of the species, of relevance to species response to climate change; available for birds, mammals and plants).*

We have used linear models (with Gaussian distribution and identity link), to relate the response to protection (as derived from the HMSC models) to the available traits, for each taxon separately.

Overall, we did not find any traits with a detectable impact on the response to protection among birds, mammals and phytoplankton, but did find a significant pattern for plants, where warm-dwelling species showed a more positive (or less negative) trend outside than inside of PAs. This finding suggests that PAs may buffer the expansion of warm-dwelling plant species under climate change. We have now introduced this result in the main text (lines 139-147), and show the model outcomes in Supplementary Material (Table S1).

Line 118-119. The statement of high statistical support needs to be revised since frequentist statistical inference does not support such a qualitative interpretation.

Thank you for this comment, which made us realize we could be more explicit about the model and how the results have been treated. We have now added further details about this in the methods (lines 382-385). Please note that here we applied a Bayesian analysis framework to estimate the species responses to protection (specifically the joint species distribution modeling framework HSMC), which unlike the frequentist inference is based on the posterior support level (based on the comment, we believe that the Reviewer might have missed our presentation of the type of approach used here.) Our approach for classifying species was based on a recent study which applied the same Bayesian framework: <https://www.nature.com/articles/s41558-022-01381-x>

Lines 129-131. This is one of the most interesting results your study brings. Is there any hypothesis explaining this?

In the revised manuscript, we have added an expanded discussion of potential reasons for our results. Our explicit hypothesis is that PAs are too small and fragmented to fully counter biodiversity loss, and the main reason for why we see a slowed-down rate of decline may be that of the remaining populations, individuals aggregate in the high-quality habitats offered by PAs (see lines 165-170). The new findings from the analysis on PA size also support this hypothesis.

Line 134. There are some typos and edition problems all over the manuscript. This may be the result of the review version, but I suggest you look at the text edition overall.

Thank you for pointing this out. In the revised version we have paid specific attention to correcting typos and editing issues.

Lines 126, 138, 140. You called those lines red and orange in different lines. I suggest you keep the same nomenclature, which, in my opinion, red fits best.

Done

Lines 166-167. This statement appears first here during my reading; however, it should be presented before. I suggest that because this new information entirely changes the interpretation of results and understanding of how PAs impact species. Also, you could explore variations on slopes between periods before and after the PAs, this could show how PAs might affect the species and consider the periods before their establishment since your data contemplate a broad period, and this feature might be interesting to evaluate impacts.

These are all valid points. However, doing a before-after comparison is not really possible with these data, as we would lose many sites and species. In fact, for plants we would lose almost all sites, mostly due to a lack of survey data derived before PA establishment. Regarding the suggestion to move the “after PA” -results before the main results, we have now carefully considered it and decided to present these results of the timing of PA designation, as well as the effect of PA size, along with the main results (see first part of the results and discussion section).

Line 183. This statement is repeated during your text. I suggest changing this to include a few sentences in the introduction and a paragraph in the results and discussion section.

This is a good suggestion, which we have implemented in the revised manuscript. In particular, we have removed the last paragraph of the discussion and incorporated the content into the other sections, thereby avoiding redundant text and shortening the section.

Lines 270-274. This assumption contrasts with the results presented between lines 165-187. Also, your sensitivity analyses only include years when PAs were established. However, my concern about this is when comparing years before and after protection and only for periods after protection. I do not think a sensitivity analysis, only considering the year when PAs were established, is sufficient to show the robustness of your assumptions. Different species and groups might have different answers to the time they are protected, and the period before protection might influence the response of these groups differently. Thus, a more profound investigation of the robustness of your approach should focus on time-series for periods after protection and not only for the year of PAs' establishment.

We believe the Reviewer may have misunderstood the way in which this analysis was done, likely because of unclear description from our part. We apologize for this previous unclarity and have now added clarification on this (Supporting Figures S4-5). We used the whole time-series including all years after PA establishment, as the Reviewer suggests to do, and not only the year of establishment.

Lines 276-299. I have some concerns about using only environmental data for matching purposes. Usually, counterfactual reasoning seeks to estimate alternative states for your causal factor of interest; here the PAs. Thus, covariates to find matching groups should rely on causal theories, which have some differences in how they approach counterfactual reasoning. However, in general,

those different theories have a common point of view, where counterfactuals should be based on a set of characteristics that influence both (i) your outcome of interest and (ii) the likelihood of establishing your intervention of interest at the specific areas they were established. In this sense, environmental variables are clearly related to the outcome and can be to the intervention; however, many other factors might influence the distribution and establishment of PAs and how protection can achieve their goals. Considering this, using only environmental variables can result in bias when estimating impacts.

Please check this paper for more about this concern: Coetzee BW, Gaston KJ. An appeal for more rigorous use of counterfactual thinking in biological conservation.

We are grateful to the Reviewer for challenging us to apply a maximally stringent and holistic approach to the counterfactual design and the matching of sites. We extensively replied above to a similar comment by yourself, and also by Reviewer 1, and provided strong evidence that indeed the variables used for matching sites and derive counterfactuals are not only relevant towards the outcome of interest, but also to the establishment of the treatment. The latter is indeed true given that we performed exact matching within the same bioclimatic zone, as latitude strongly correlates with PA establishment in Finland. As you also pointed out earlier, most of the largest PAs in Finland are located in the northernmost zone where land is less productive and valuable for anthropogenic use. We however fully agree that unobserved confounders are likely relevant, and we discuss this caveat in the manuscript (lines 340-347).

Reviewer #3 (Remarks to the Author):

Comments to the authors of 'Mixed effects of protected areas on terrestrial and freshwater biodiversity' (manuscript #NCOMMS-23-04671).

In this study, the authors evaluate the effectiveness of protected areas (PAs) in promoting the occurrence (i.e. rates of presences and absences) of 638 species in the boreal ecosystems of Finland through time (from the 70s or the 80s depending on the group) and across four clades (i.e. mammals, birds, vascular plants and freshwater phytoplankton). Based on extensive monitoring surveys that document changes in species occurrences through time, the authors use the now established Hierarchical Modelling of Species Communities (HMSC) framework to model the effect of conservation status (i.e. protected vs. unprotected) and time (i.e. year) on the distribution of occurrence probability at the species-level. The HMSC framework accounts for differences in sampling effort between clades as well as spatio-temporal autocorrelation. Survey sites located close enough to an official PA (according to a spatial buffer) are assumed to be under its influence. Sites classified as unprotected were not randomly sampled through the study area. Indeed, the authors chose unprotected sites displaying similar landscapes/environmental conditions as the PAs. This is a nice and important feature of the study as the trends modelled are thus independent of species-level differences in niches/ environmental preferences.

The authors find that most of the sites likely influenced by PAs display no detectable impact on species occurrences compared to sites not influenced by PAs, with some slight differences across clades: 90% and 92% of the phytoplankton and vascular plant species, respectively, show no detectable temporal trend in occurrences, whereas mammals and birds show higher proportions of detectable effects (78% and 69%, respectively). When the effects of PAs establishment are statistically supported, they are either positive or negative with no clear advantage towards positive effects. The authors also show that the presences of birds and phytoplankton species are promoted once PAs are established, thus supporting the view that PAs are not ineffective neither. Yet, this was not the case for mammals and a large variety of responses to PAs establishment were observed within clades. Consequently, the authors conclude that PAs can have a positive

impact on the occurrence of species/groups but that there are not sufficient to protect entire communities. This study contributes to the ongoing and critical debate on the effectiveness of PAs as useful units for biodiversity management and conservation, and show how species-specific responses must be accounted for to achieve efficient community-level conservation.

I find the present submission to be of fine quality overall. The text is very pleasant to read since the language is of prime quality, the study is very well motivated (i.e. the introduction is excellent!) and the graphs are clear and convey the main results effectively. The authors made the effort to evaluate the sensitivity of their main findings to some of the key choices made in the numerical framework (i.e. year of PAs creation, spatial buffer around the PAs, spatial variability in PAs size, method used to match PAs with unprotected areas) and all this information is summarized in a comprehensive suite of Supplementary Materials. I do not see major flaws in the data analyses and their interpretation. The work carried out by the authors does support their conclusions in my opinion. The methodology is well explained and exhaustive enough to be reproduced (provided the authors make their data and R codes publicly available).

Thank you for your positive comments and your appreciation of our work. The code and data are now available in the Figshare repository, see link in the manuscript.

I understand that submissions to Nature Communications must remain quite short. However, I think the Discussion of the paper needs to include additional elements to better put the present results into perspective (see comments #1 to #4 below). I find that the last 10-15 lines or so of the main text are way less interesting/relevant and could be replaced by more insightful elements centered around the points I am highlighting below. To do so, the authors may need to run some additional numerical analyses but I am confident they will be able to manage these. To conclude, provided the authors address my comments below, I would recommend this study for publication.

We are most grateful to the Reviewer for their positive and encouraging words. We have now addressed all the concerns highlighted, and have revised the discussion accordingly (lines 197-221).

1. PAs effectiveness for conserving the abundance/biomass of the species/clades ?

At line 254, the authors mention that all quantitative abundance values were converted to presence-absence to run the JSDMs that are part of the HMSC framework. Is this because species counts are not comparable across the four clades or is because the JSDMs cannot handle abundance data? Please clarify. Also please document the number (in relative contribution for instance) of records/taxa that were discarded because they were not identified at the species-level. I think readers should know how representative the final dataset is compared to the full survey data. How many taxa/sub-groups are lost because of this filter?

I am surprised that the authors do not mention the lack of abundance data as a critical limitation/assumption of their study. Evidently, species can be present in varying levels abundances. For ecosystem management and for evaluating ecosystem functioning, species abundance/biomass should be more informative than occurrence. PAs could be misclassified as effective if they only manage to retain species in very low quantities (i.e. the populations of the species are barely surviving). This is something that should definitely be mentioned and discussed in the main text (see Coetzee 2016, <https://doi.org/10.1007/s10531-016-1235-2>; Johnston et al., 2013 <https://doi.org/10.1038/nclimate2035>; Gray et al., 2016 <https://doi.org/10.1038/ncomms12306>), especially for a journal with such a wide audience as Nat. Comms.

I understand how the different surveys can be difficult to integrate as their methods can be hardly inter-comparable. Yet, I wonder if parts of the present dataset could not be used to assess the

impact of PAs on abundance rather than occurrences (i.e. not necessarily through JSDMs). Within clades, have the authors tried to examine time series of species abundances? If possible, I would strongly encourage the authors to try to integrate this in their manuscript, at least to discuss this major caveat.

Regarding the data filtering step, this did not result in the loss of any taxa/groups, but only in the exclusion of a minority of observations from individuals that had not been identified to the species level in the monitoring data.

The comment regarding abundance is a relevant one. HMSC could in theory handle abundance data, but the issue is indeed that the counts will not be directly comparable among taxa due to differences in how the taxon-specific abundance was scored. While this is not a major impediment, as we run one model per each taxon separately, we did attempt fitting of a hurdle model on abundance data (i.e. abundance conditional on presence). Unfortunately, these models of abundance conditional on presence simply did not converge, likely caused by some violations of joint distribution assumptions and/or computational demands for estimating abundance values across large community matrices (i.e. tens to hundreds of species across hundreds to thousands of sites x many years). We now introduce this lack of abundance results as a caveat in the discussion (lines 208-210).

2. PAs effectiveness for conserving species diversity (alpha and beta, phylogenetic) and functions (functional diversity indices) ?

Similarly, I wonder if the authors cannot integrate diversity indices in their study to evaluate the performance of the PAs for conserving biodiversity rather than species-levels occurrences. Would not these indicators be better suited to assess community-level patterns? Have the authors tried to look at the spatio-temporal dynamics of indices such as Shannon-Wiener or Simpson's (alpha diversity) or Bray-Curtis distances (or Jaccard's or Sørensen's; for beta diversity)? Similarly, could the efficiency of the Finnish PA network vary across the phylogeny or the functional traits of the species? Considering that a wealth of phylogenies and functional traits compilations exist for the clades studied (except maybe for the phytoplankton), I wonder if the authors could not integrate indices of phylogenetic diversity or functional diversity in their analyses? Or maybe they are considering integrating those in follow-up studies? If the goal of the authors is indeed to examine the effects of PAs on Biodiversity (i.e. the title of the study), I feel like they should go beyond occurrences and integrate more dimensions of Biodiversity.

We agree with the Reviewer that assessing how PAs affect other aspects of diversity is very relevant and would provide important insights into community-level patterns. However, that would stray well beyond the objectives of the current study. Given that our aim is to address the impact of PAs on species, then the relevant metrics are the occurrence, abundance and distribution of species – of which diversity metrics will be poorly reflective. Thank you for suggesting to include new analyses to test a phylogenetic or functional signal in species responses to protection (see reply to similar comment above). We have now included analyses where we test the effect of functional traits on species response (lines 138-147), where we reveal an effect of thermal niche for plants (Table S1). We have also added tests for phylogenetic signals in the responses which we visually represent by mapping species responses onto the phylogenetic tree of each taxon (new Figs.22-25 in the supplementary material); overall we found no phylogenetic structuring for the effect of PAs. These analyses are clearly framed on the question of how species respond in occurrence probability to PAs, and could thereby be included in the study without straying off topic.

3. How can functional traits help explain differences in PAs effectiveness across clades ? Are there any trends within clades (e.g. Order-levels? Family-level? Genus-level?) ?

Along the same lines as above, I strongly encourage the authors to better discuss the potential effects of species' traits (or clade-level traits) on the observed differences in trends? The authors find that the rates of occurrence increase post-PA establishment for birds and phytoplankton, but not for vascular plants and mammals. This is quite interesting as birds and plankton could be viewed as groups that are less subject to dispersal limitations than vascular plants and mammals. Overall, the four clades display major differences in life histories, life cycle length, physiologies, thermoregulation (ectothermy vs. endothermy), dispersal capabilities etc. How can those explain differences between clades? What about intra-clade variability in traits? What biases are introduced when comparing phytoplankton that passively live in interconnected lakes (how well are those lakes connected in Finland?) and birds that can fly miles to nest in the habitat that they find suitable? Furthermore, could the authors did try to interpret/explain the high level of within clade variability (Figures 3 & 4) through species specific traits? Among the positively/negatively impacted taxa, are there specific sets of traits that could explain the positive/negative influence of the surrounding PA? Are there any intra-clade patterns from a taxonomic point of view (family- level, order-level etc.)?

The authors perform JSDMs but do not seem to make use of all the information those can give. Following Pollock et al. (2014), « JSDMs [...] allow decomposition of species co-occurrence patterns into components describing shared environmental responses and residual patterns of co-occurrence ». Therefore, I was wondering if the authors could not identify sets of species (within clades) that display co-occurrence patterns driven by shared environmental responses, and test whether those sets display similar temporal dynamics in PAs (e.g. same responses in Figures 3 & 4).

We thank the Reviewer for this suggestion, which has greatly improved our insights on the effects of protection and species responses, and we now present a trait-based analysis as suggested (see also our reply above).

While we find the many lines of further exploration offered by the Reviewer most interesting in themselves, we are highly aware that a brief paper in a journal like Nature Communications will call for a very well-defined focus. For this reason, we must stress that our current analysis is aimed at quantifying the effect of PAs on species level occurrences, rather than species co-occurrence, which is conceptually a very different question and would require a separate study with an amended study design to be addressed. However, this is something that could be conducted in the future by our research team, and we appreciate the idea provided by the Reviewer.

4. Is there an effect of PA category on species occurrences? Can the features and enforcement levels of PAs help explain differences between and within clades ?

Another major caveat of the study that the authors should better discuss in the main text is the fact that not all PAs are equal. As the authors must be aware of, the IUCN defines no less than seven categories of PAs, from « strict reserves » to « areas with sustainable use of resources ». First, I encourage the authors to mention the types of PAs that are present in their study area (i.e. the relative contribution of each IUCN PA category; a supplementary plot could be shown; or color the points of Figure 1 as a function of IUCN status? Which makes me think that there could be an interaction between space and the status of the PA)? Second, could these categories be encoded as hard/random effects in the HMSC? I expect the level of protection/ enforcement to have a major impact on the efficiency of the PAs. Therefore, could the positive trends be driven by PAs characterized by more stringent levels of conservation? Would it also make sense to make the spatial buffer around the PAs vary also as a function of conservation status (i.e. the level of PA

influence could increase with the level of conservation?)? Overall, I strongly suggest to better discuss this in the manuscript (unless all the PAs of the study area have the same status?).

We are again grateful for these relevant suggestions, which we have now added to the revised manuscript. Indeed we used PAs belonging to different IUCN protection categories, an approach that follows what was done in several recent studies (e.g. Xu et al., 2023; Terraube et al., 2020; Montesino Pouzols et al., 2014). While not all PAs in Finland have been assigned an IUCN category (23% out of the total 9000+ PAs do not currently have an assigned IUCN category status), thus precluding a robust analysis on this potential effect, we have now added an additional sensitivity analysis to verify the robustness of the main results to variation in PA IUCN category level based on the available data. Specifically, we re-ran the analysis using the subset of PAs belonging to IUCN category I to IV, although realizing that an unknown proportion of the unassigned PAs could in reality also belong to these categories. Categories I to IV are the ones that require the strongest protection levels, e.g. even sustainable resource use is not allowed (as compared to IUCN categories V and VI where this is allowed). In addition, we have also added another analysis to explicitly test the effect of PA size in modulating the effect of protection on species occurrences. The results of the IUCN PA category analyses qualitatively align with the main results, whereas the results of the PA size analysis reveal that when only the largest fraction of PA sizes is considered, the effect of PAs becomes more positive for mammals, plants and phytoplankton, but not for birds. We have now included a discussion of these results in the main text, while we present the results of the additional analyzes in the supporting material (Supplementary Figs. 6-7).

Xu, W.B., Blowes, S.A., Brambilla, V. et al. Regional occupancy increases for widespread species but decreases for narrowly distributed species in metacommunity time series. Nature Communications 14, 1463 (2023).

Terraube, J., Van doninck, J., Helle, P. et al. Assessing the effectiveness of a national protected area network for carnivore conservation. Nature Communications 11, 2957 (2020).

Montesino Pouzols, F., Toivonen, T., Di Minin, E. et al. Global protected area expansion is compromised by projected land-use and parochialism. Nature 516, 383–386 (2014).

5. Selection of environmental covariates (more minor comment).

The authors rely on the CLC data to classify survey sites according to their environmental similarity with the PAs. Four additional parameters (lake size, lake water color, phosphorus concentration and nitrogen to phosphorus ratio) were included for freshwater phytoplankton. I was wondering why the authors did not choose to include lake surface temperature for phytoplankton (and vascular plants) as well? There is now widespread evidence that temperature controls the physiology, distribution and production of freshwater phytoplankton species. I find that this is a key factor for measuring the similarity of survey sites for phytoplankton and plants, as not all species can occur between areas that vary sometimes by even $< 5^{\circ}\text{C}$.

Thank you for this suggestion. As for the environmental covariates for matching phytoplankton sites, we used those that, in a recent paper on phytoplankton occurrence in Finland, had the highest influence on phytoplankton occurrence (Weigel et al. 2023). As temperature explained a very negligible fraction of the variance in phytoplankton occurrence, we did not include it here. Note that we also included water color, P and P/N ratio as water parameter covariates for the matching, and these likely also capture, at least partly, the variation in temperature.

Weigel, Benjamin, et al. "Macrosystem community change in lake phytoplankton and its implications for diversity and function." Global Ecology and Biogeography 32.2 (2023): 295-309.

Code availability: to abide the FAIR principles for data sharing and stewardship, I would ask the authors to make their data and R codes already available on an online repository such as Zenodo or GitHub.

The data and code used in the analysis are deposited in figshare repository and will be made public upon acceptance.

I hope the authors will find my comments useful and that they will use them to improve their interesting study.

We are grateful for your time and comments, which we found very constructive and useful for further improving our work.

Yours faithfully,

Reviewer #4 (Remarks to the Author):

Mixed effects of protected areas on terrestrial and freshwater biodiversity

Submitted to: Nature Communications

Authors: Santangeli et al.

General comments:

Thank you for the opportunity to review this manuscript. Although I do appreciate the effort and analysis put forth by the authors, I do not think the contribution warrants publication in the journal Nature Communications. I do think, however, that the work is publishable in a different outlet.

I provide comments below that explains my decision, but I think the "impact" and contribution of this work was best summarized by the authors in the last sentence of their Abstract that states "Better monitoring and robust assessment of their effectiveness are fundamental to leverage the benefit of protected areas and to bend the curve of biodiversity loss." I think this statement is an accurate reflection of the analysis and results, but unfortunately, if the main take-home message is that we need to collect more ("better") data, then it is my opinion that the contribution and impact of this work is not suitable for this outlet. The main results of whether or not a species showed a "positive" or "negative" response to Protected Area

(PA) was not very compelling for the following reasons:

I found the categorization of "positive", "negative", and "no effect" difficult to interpret ecologically. For example, a PA had a negative effect if there were slower (less positive increase) rates of increase in occurrence probability compared to outside of PAs. Without species level information and a better discussion of effect sizes, this "negative" effect is really difficult to assess or even to know if it is in fact "negative" from an ecological perspective. It may be that a less rapid increase in occurrence probability over time in a PA is a "positive" ecological effect given the species and effect size. The authors acknowledge this starting line 155, but I find the this issue is severe enough that it really limits the usefulness of the assessment. Also, the overall effectiveness of PAs for most species was "no detectable effect" - again, this may be a "positive" effect in some cases.

For common species showing no negative trend outside a PA, perhaps having no detectable trend in the PA is a good (positive) outcome? These "no detectable effect"

species are the majority, but there is little additional investigation or discussion about them.

We thank the Reviewer for their critical comments, which have led us to improve our work and expand the results presented previously. We have robustly estimated species responses to protection for hundreds of species across different taxonomic groups, highlighting that most species are largely unaffected by protection, while a minority shows positive or negative effects of protection. We agree that the interpretation of the PA effect on species is complex. We believe that our criteria for estimating these effects on species occurrences are both justified and follow best-practice as outlined recently in Wauchope et al. 2021 and 2022. We apologize for any lack of clarity in the previous version, and have now invested further effort in clarifying our approach and justifying the criteria. The categorization of the responses to protection followed the categories that have been recently suggested as a gold standard for impact evaluation (Wauchope et al. 2022). We are aware that species responses to PAs fall along a continuum, and this is why we present this variation in the former Figure 4 (now Figure 3). We now address the ecological implications of the different response types in the discussion, e.g. the potentially higher relevance of finding a positive impact on high latitude species in Finland (lines 197-210). Finally, we have now added a trait-based analysis aimed at better understanding what features make species prone to responding in a specific way to protection.

*Wauchope, H. S., Amano, T., Geldmann, J., Johnston, A., Simmons, B. I., Sutherland, W. J., & Jones, J. P. (2021). Evaluating impact using time-series data. *Trends in Ecology & Evolution*, 36(3), 196-205.*

*Wauchope, H. S., Jones, J. P., Geldmann, J., Simmons, B. I., Amano, T., Blanco, D. E., ... & Sutherland, W. J. (2022). Protected areas have a mixed impact on waterbirds, but management helps. *Nature*, 605(7908), 103-107.*

Related to the above point - there is no discussion of what types of species within the broad taxonomic groupings may or may not be responding in some way to PAs - other than designating species as "threatened" or "non-threatened", were there certain species traits that corresponded to type of PA response (assuming you can interpret those responses in a meaningful way)? I think some treatment of species-level characteristics is needed to make these types of analyses useful for practitioners. For example, the authors make the argument that there is an urgent need for this type of analysis, but without some investigation at the species-level, I question its utility to actually inform the management, use, etc. of PAs. E.g., just knowing the % "birds" that showed a "positive" response to PA does not seem that useful to guide management and climate adaptation strategies.

We thank the Reviewer for challenging us to find broader patterns that can help explain species' responses. We have now introduced extensive trait-based analyses and results (see also response to a similar question above) and find that they do not help explain which species tend to respond positively or negatively to protection, except for thermal preference for plants.

Generally a better treatment, illustration, and discussion of effect sizes, if they are ecologically relevant (not just statistically significant), is needed. The authors provide the protected area effect in Figure 3, but the logit-scale interaction term is not easily interpreted for ecological relevancy.

As mentioned in the above response to a similar comment, interpreting the response to PAs is complex, and we detail above how we addressed this, also with the trait analysis, and by presenting the response as continuous value (former figure 4, now figure 2) rather than as categories of responses. Note that here we have used the same approach to classify the species into positive, negative or no effect of protection from across the four groups, so that the results are comparable among them.

Other comments:

Abstract: Line 31 (and throughout) - what are the effect sizes you are talking about here and are they ecologically significant/meaningful. "Slower rates of decline" cannot be interpreted in an ecologically meaningful way.

We have addressed this comment above in reply to a similar concern. After performing the matching of protected and unprotected sites, the effect sizes given here are indeed comparable between species. Whether they are ecologically significant will again be context and species specific. Pragmatically, and conceptually speaking, though, the results presented should be effective in assessing the impact of protected areas, in much the same way as suggested by the framework presented by Wauchope et al. 2021 as well as in their empirical evaluation using waterbirds published in Nature in 2022 (references above).

Introduction: Line 37 - "ecological impacts" on what? Are you just talking generally about global-scale impacts of humans or specific impacts on certain systems?

This is a general statement related to the Anthropocene and anthropogenic impacts on biodiversity. We have now defined it more accurately, by replacing "impacts" with "appropriation of nature", which is more specific and compelling.

Line 50: I would rephrase ". . . robust PA impact evaluation is problematic" to ". . . assessment of PAs is problematic

Modified as suggested.

Line 50: Some brief discussion of how PAs are evaluated, if not using a counterfactual approach, is warranted.

We now added a sentence outlining this (lines 56-59).

Line 54: I would avoid using terms like "even worse" (worse than what?), "easily" (line 56), "urgent" (line 58), "striking" (line 85), etc. The use of these terms are vague, not quantifiable, and sometimes give the impression of trying to increase the visibility and contribution of the work. For example, the sentence on line 85 that states the lack of protection effect was particularly striking for plants and phytoplankton - I did not find these results particularly striking given the ecology of these organisms and the reliance on occurrence data.

We have now revised the text to remove any vague or subjective terms, as suggested.

Line 56: This analysis is focused on PA effectiveness in terms of preserving biodiversity, but they serve other functions (e.g., reducing carbon fluxes to the atmosphere) that are considered during the development of PAs, but this is not touched on in this paper. I think a sentence of two of the broader ecological importance of PAs is warranted.

Thank you, we have now added the point about the role of PAs as also preserving ecosystem services and functions (lines 206-208), as suggested.

Results and Discussion: Line 82: I think you need to start the results with a discussion of the what the trends were in PAs and outside PAs before you jump into if species were "protected". Again, I find the lack of discussion of effect sizes makes it difficult to assess importance of the work. What were ranges in the probability of occurrence over time for different species?

See reply to your similar comment above. We have now improved our discussion concerning the interpretation of the results. We believe the results are clear and direct as they are, they are comparable among the four taxonomic groups, and also follow the protocol showcased in a recent similar study: Wauchope et al. 2021 Nature. We further stress that we presented the estimate of beta (which is at the scale of the linear predictor, hence with a probit link to the data scale) for the effect size of the protected area variable.

Figure 3: caption and figure - it states ". . . where darker colors indicate higher statistical support for the effect." I am left to assume that this is the posterior probability of a positive or negative effect, but that should be stated, if so. Alternatively, to improve readability of this figure, just shade the ones with Prob>90% to align with your aforementioned cutoff. The use of a probability cutoff above and a gradient here does not help readability (although I appreciate not relying on a cutoff of artificial significance

generally).

We are glad that the referee appreciates our decision to present the results in a most direct and open way, that is showing the full gradient of effects. In this light, we also moved the former figure to the Supplementary Materials.

Figure 4: What about no effect in or out of PAs, which are the majority of the results. Again, I think it is potentially inaccurate to suggest that "no effect" is not potential "protection" for some species - especially if the trends in and out of the PA are relatively flat (it may not be protecting better than sites outside the PA, but still may be protective in the broader sense of preserving habitat and having longer term benefits by being a PA).

This is a very valid point, and we now discuss it in the revised manuscript (lines 206 - 210).

It seems the authors have the expectation that PAs should be "protective" for all species for the PA to be considered effective. Would we really expect all species to increase in occurrence probability in a PA over time? This seems like a very high standard and one which will lead to the results presented - that PAs are perhaps not great at preserving biodiversity. I understand the authors caveat this finding, but nonetheless it is one that people will likely take away from this paper.

In interpreting the results on responses to protection, we have closely followed the framework of Wauchope et al. 2022 Nature. We now present a new trait-based analysis which casts more light on what groups of species benefit from PAs, and what groups do not. Please also see our reply to your previous point and how we addressed it.

Line 149 states "Generally, only the cases where PAs actually mitigate or reverse declines can be seen as unequivocally positive conservation outcomes." I think this statement oversteps a bit - what can be viewed as a positive conservation outcome is dependent on the goals and objectives of the PA and might be species specific. I think this statement simplifies the equation too much,

Please see our response to your comment above

Lines 171 - 177: Again, the authors acknowledge that the main contribution of their work is to suggest that we need "rigorous assessment" - a finding that I don't think warrants publication in Nature Communications.

We kindly disagree to this interpretation, in fact also the three other Reviewers found the study interesting and relevant. To the best of our knowledge, this is the first time that a rigorous assessment employing a counterfactual approach is implemented utilizing longitudinal wildlife survey data including very different taxonomic groups, with species from terrestrial to freshwater ecosystems. A study by Terraube et al. (2020 in Nature Communications) used a similar counterfactual approach, but included only four carnivore species. Even the excellent recent study by Wauchope et al. 2022 published in Nature did not have such a diversity of taxa (was focused on waterbirds, albeit a species-rich group) that we had. We show that species responses to protection are not universal, but that species in general remain present for longer inside than outside of PAs. With the additional species-trait analysis results, and those on protected area size and timing, we are able to generate even stronger conclusions and widen the already large implications of this study. We thus thank the Reviewer for their constructive suggestions, and we believe our work is a relevant and timely contribution for improving conservation, providing key insights towards the implementation of the post-2020 Biodiversity Framework, and for further contributing to advance the science of impact evaluation

Terraube, J., Van Doninck, J., Helle, P., & Cabeza, M. (2020). Assessing the effectiveness of a national protected area network for carnivore conservation. Nature Communications, 11(1), 2957.

Wauchope, H. S., Jones, J. P., Geldmann, J., Simmons, B. I., Amano, T., Blanco, D. E., ... & Sutherland, W. J. (2022). Protected areas have a mixed impact on waterbirds, but management helps. Nature, 605(7908), 103-107.

Line 181 states ". . . for improving conservation practice and avert the biodiversity crisis" - the crisis is here my friends, there is no averting it. We can try to minimize its magnitude, but it will not be averted.

We sadly agree. We now replaced that whole sentence. We now talk about mitigating rather than averting.

Statistical analysis: I appreciate the thoughtfulness of the treatment of the data, the analysis, and the assessments of results to assumptions and model structure and data decisions (i.e., the sensitivity analyses).

Thank you.

Minor comments:

Supplement tables 1 and 2: I know this is a regional preference, but it might be better presented to use decimal points throughout instead of decimal commas as in Table S1.

We replaced all decimal commas with points in all tables, as suggested.

Thanks for your time and effort in reviewing our work and providing feedback. Your comments helped us to improve this work even further.

Reviewers' Comments:

Reviewer #1:

Remarks to the Author:

The response to review is excellent. They have provided really valuable explanation to justify their approach (eg the use of joint species distribution models). I learnt a lot from this explanation.

They have done quite substantial new work eg the work assessing the impact of functional traits on species responses in reply to reviewer 3 and 4, and the checks (for the subset of Pas with an IUCN category) of whether effect of protection varies with UCN status.

I see why the reviewer 2 referred to the counterfactuals as 'weak'. But while they are not what would be ideal, this is somewhat inevitable in such observation studies). The author's response on this point is very strong. I feel the paper is clear enough about the caveats and given the value of the data set and the effort they have put into the design to make the inference as strong as possible, suggests it should be published.

I disagree with the comments of reviewer 4 who feels that because the conclusions are partly that the data collected on protected areas doesn't allow for a robust evaluation means the paper isn't interesting.

This reviewer also seems to also have rather unrealistic expectations. The value of this paper is the broad taxonomic coverage and the large number of species for which data is presented. Inevitably results cannot be presented in the context of goals for specific species at specific sites. While such an approach would have value, and it is clearly true that the way data is analysed and interpreted here is only part of the story, it is certainly ambitious enough and interesting enough to warrant publication as will be a very valuable addition to a fast moving literature on conservation impact evaluation.

Reviewer #2:

Remarks to the Author:

I enjoyed reading the review manuscript. Therefore, I suggest publishing it without making any further critical changes. However, I recommend conducting an editorial review before publishing to improve its quality.

After reading the review letter and the new version, I am convinced that the approach and evidence are worthy of publication, especially considering the novelty of investigating the impacts of PA's on direct biodiversity units. The new changes and additions, especially for counterfactual estimations, are welcome and adequately explained by the authors. Additionally, the species-level traits and conservation status contribute to a valuable discussion about the effectiveness of PAs. Overall, I consider the text relevant and well-written, and I hope the journal accepts its publication. I only have a few concerns regarding the editing, which are mentioned in the minor comments below.

Edition concerns:

I suggest editing your figures in the main manuscript to make them look cleaner. For example, the X-axis on both figures 2 and 3 have different styles, with some showing dashed lines while others have solid lines. Additionally, the space between the legend and the axis could be larger. For the figure 3 legend, I suggest adding opening parentheses when you cite each main finding. For example, "observed through 1) alleviating rates of decline" could be "observed through (1) alleviating rates of decline." Regarding the supplementary material, the distance between the legend and the border in Figure 1 looks odd. Also, some figures in the supplementary material appear to be of low definition.

Reviewer #3:

Remarks to the Author:

Dear Authors,

I am quite satisfied with the way my previous comments were addressed by the authors. The text was improved in the places where it needed to (the last paragraph, and the discussion). The

authors carried out additional numerical analyses (reported in the main text and the supplementary material) to try to incorporate the effects of PA category and species traits in their analyses, as suggested by other reviewers and myself. I wish they had incorporated more ecological traits than "just" body size and temperature preference (IUCN status is not really a functional trait associated with an ecological function). But I understand that the amount of traits that can be realistically integrated in this study scales negatively with the amount of species and groups studied. I assume that integrating more functional traits would require much more time from the authors, which may be outside the scope of this submission. Scope-wise, I also wished the authors had integrated more community-level Biodiversity indices (i.e., my previous comments). But maybe that would not fit the scope of the study. I will let the editors decide on that.

All in all, I am happy to see that the authors found my comments helpful and I truly hope they will take my suggestions in a follow-up study. If the other (and more critical) reviewers are happy with the revised manuscript, than I am happy to see it published in Nat. Comms too. I am not sure I agree with Reviewer #4 on the fact that this study is not impactful enough to be published in Nat. Comms. I am convinced by the authors' point that similar and less broad studies have been published in high profile journals in the near past. I like that this study provides a nuanced perspective on the impact of PAs on species conservation. Hopefully, a similar study can be achieved on the European or continental scales in the future.

Best regards,

Reviewer #4:
Remarks to the Author:

**Mixed effects of protected areas on terrestrial and freshwater biodiversity –
Revision 1**

Submitted to: *Nature Communications*
Authors: Santangeli et al.

General comments:

The authors have done a nice job revising and addressing reviewer concerns. The paper reads much better now and represents an important contribution to our understanding of the role of protected areas as the planet attempts to develop strategies to address the biodiversity crisis.

Minor comments:

- Minor comment : Line 31, consider changing “via” to “through”
- Line 34-35: Should the sentence read “. . . network can partly contribute to slow down declines *in occupancy rates*, but . . . ?
- Lines 41-45: Is this a paragraph? It appears so, but not totally clear since indents are not used. If it is, I would recommend not having two sentence paragraphs.
- There is an extra space after the word “same” on line 73
- Figure 1: the unprotected sites (white circles) are a bit hard to see against the light gray background of the polygon. I suggest making the gray polygon a little darker or choosing a color other than white for the unprotected sites to increase contrast. You can also use some transparency on the points to reduce the visual effects of point overlap.
- Line 122: Add comma after 14%
- There are cases where you use a comma after *i.e.*, and times when you don't. I suggest you always use *i.e.*, and not *i.e.*

Response letter.

Authors' replies in italic font

Reviewer #1 (Remarks to the Author):

The response to review is excellent. They have provided really valuable explanation to justify their approach (eg the use of joint species distribution models). I learnt a lot from this explanation.

They have done quite substantial new work eg the work assessing the impact of functional traits on species responses in reply to reviewer 3 and 4, and the checks (for the subset of Pas with an IUCN category) of whether effect of protection varies with UCN status.

I see why the reviewer 2 referred to the counterfactuals as 'weak'. But while they are not what would be ideal, this is somewhat inevitable in such observation studies). The author's response on this point is very strong. I feel the paper is clear enough about the caveats and given the value of the data set and the effort they have put into the design to make the inference as strong as possible, suggests it should be published.

I disagree with the comments of reviewer 4 who feels that because the conclusions are partly that the data collected on protected areas doesn't allow for a robust evaluation means the paper isn't interesting.

This reviewer also seems to also have rather unrealistic expectations. The value of this paper is the broad taxonomic coverage and the large number of species for which data is presented. Inevitably results cannot be presented in the context of goals for specific species at specific sites. While such an approach would have value, and it is clearly true that the way data is analysed and interpreted here is only part of the story, it is certainly ambitious enough and interesting enough to warrant publication as will be a very valuable addition to a fast moving literature on conservation impact evaluation.

Thanks for your valuable and constructive comments, and for the high appreciation of our work.

Reviewer #2 (Remarks to the Author):

I enjoyed reading the review manuscript. Therefore, I suggest publishing it without making any further critical changes. However, I recommend conducting an editorial review before publishing to improve its quality.

After reading the review letter and the new version, I am convinced that the approach and evidence are worthy of publication, especially considering the novelty of investigating the impacts of PA's on direct biodiversity units. The new changes and additions, especially for counterfactual estimations, are welcome and adequately explained by the authors. Additionally, the species-level traits and conservation status contribute to a valuable discussion about the effectiveness of PAs. Overall, I consider the text relevant and well-written, and I hope the journal accepts its publication. I only have a few concerns regarding the editing, which are mentioned in the minor comments below.

Edition concerns:

I suggest editing your figures in the main manuscript to make them look cleaner. For example, the X-axis on both figures 2 and 3 have different styles, with some showing dashed lines while others have solid lines.

This comment stems from a possible misreading of the x-axis, not being a conventional line, but rather representing tick marks, one per species. As the x-axis is indeed categorical, with each bar of the plot representing the pattern for each species, we feel using species-level tick marks is most appropriate. Of course this may cause the tick marks to clutter for the most species rich taxa plots, like phytoplankton, as the referee pointed. We clarified now in the caption of this figure what the x-axis really is, to be very explicit with the reader.

Additionally, the space between the legend and the axis could be larger.

This is merely an individual aesthetic preference, which we understand, but again we feel the figure is not heavy enough or cluttered to necessitate spreading the items in it any further. However if the editors prefer otherwise we are happy to apply modifications on it.

For the figure 3 legend, I suggest adding opening parentheses when you cite each main finding. For example, "observed through 1) alleviating rates of decline" could be "observed through (1) alleviating rates of decline."

Done.

Regarding the supplementary material, the distance between the legend and the border in Figure 1 looks odd. Also, some figures in the supplementary material appear to be of low definition.

Regarding Figure S1, the distance mentioned is inevitably there as we are showing the full range of potential values for the patterns, from 0 to 100% for both negative and positive effects. We cut add a break point to cut that space but we feel it is more clear to the reader to show the figure as it is. Also, as this is included in the supplementary material, it has no space constraints, therefore we feel it is most appropriate to present it as it is. Again, this is a largely aesthetic decision, and if the editors prefer otherwise we are happy to apply modifications on it.

Thanks for your valuable and constructive comments, and for the high appreciation of our work.

Reviewer #3 (Remarks to the Author):

Dear Authors,

I am quite satisfied with the way my previous comments were addressed by the authors. The text was improved in the places where it needed to (the last paragraph, and the discussion). The authors carried out additional numerical analyses (reported in the main text and the supplementary material) to try to incorporate the effects of PA category and species traits in their analyses, as suggested by other reviewers and myself. I wish they had incorporated more ecological traits than "just" body

size and temperature preference (IUCN status is not really a functional trait associated with an ecological function). But I understand that the amount of traits that can be realistically integrated in this study scales negatively with the amount of species and groups studied. I assume that integrating more functional traits would require much more time from the authors, which may be outside the scope of this submission. Scope-wise, I also wished the authors had integrated more community-level Biodiversity indices (i.e., my previous comments). But maybe that would not fit the scope of the study. I will let the editors decide on that.

All in all, I am happy to see that the authors found my comments helpful and I truly hope they will take my suggestions in a follow-up study. If the other (and more critical) reviewers are happy with the revised manuscript, than I am happy to see it published in Nat. Comms too. I am not sure I agree with Reviewer #4 on the fact that this study is not impactful enough to be published in Nat. Comms. I am convinced by the authors' point that similar and less broad studies have been published in high profile journals in the near past. I like that this study provides a nuanced perspective on the impact of PAs on species conservation. Hopefully, a similar study can be achieved on the European or continental scales in the future.

Best regards,

Thanks for your valuable and constructive comments, and for the high appreciation of our work. We fully agree that deeper investigation on the functional traits that mediate species responses to protection are important and relevant. While out of the scope, and space capacity, of this study, we certainly will keep this a potential research topic in the near future.

Reviewer #4 (Remarks to the Author):

General comments:

The authors have done a nice job revising and addressing reviewer concerns. The paper reads much better now and represents an important contribution to our understanding of the role of protected areas as the planet attempts to develop strategies to address the biodiversity crisis.

Minor comments:

Minor comment : Line 31, consider changing "via" to "through"

done

Line 34-35: Should the sentence read ". . . network can partly contribute to slow down declines in occupancy rates, but . . . ?

done

Lines 41-45: Is this a paragraph? It appears so, but not totally clear since indents are not used. If it is, I would recommend not having two sentence paragraphs.

Indeed this is a paragraph, and we feel it is OK to leave it as is, short and focused, as it nicely opens the field in terms of what is the problem (biodiversity crisis), one of the solutions (protected areas), and how PAs have been assessed so far. The next paragraph is clearly separated in its topic that it would not make sense to merge these 2 into one long and heterogeneous single paragraph.

There is an extra space after the word "same" on line 73

done

Figure 1: the unprotected sites (white circles) are a bit hard to see against the light gray background of the polygon. I suggest making the gray polygon a little darker or choosing a color other than white for the unprotected sites to increase contrast. You can also use some transparency on the points to reduce the visual effects of point overlap.

We tried increase the darkness of the grey polygon but the overall appearance worsened, looks heavier. Same for transparency, which made the points less visible. As these maps are only meant to show the broad coverage of the data across the country, we feel they are clear enough as they are to fully convey the main message. Again, this is a largely aesthetic decision, and if the editors prefer otherwise we are happy to apply modifications on it.

Line 122: Add comma after 14%

done

There are cases where you use a comma after i.e., and times when you don't. I suggest you always use i.e., and not i.e.

done

Thanks for your valuable and constructive comments, and for the high appreciation of our work.